# Prevention of Venous Thromboembolism in 2020 and Beyond

**DOI:** 10.3390/jcm9082467

**Published:** 2020-08-01

**Authors:** Matthew Nicholson, Noel Chan, Vinai Bhagirath, Jeffrey Ginsberg

**Affiliations:** Division of Hematology and Thromboembolism, Department of Medicine, McMaster University, Hamilton, ON L8N 3Z5, Canada; Noel.Chan@taari.ca (N.C.); bhagiv@mcmaster.ca (V.B.); ginsbrgj@mcmaster.ca (J.G.)

**Keywords:** VTE, PE, DVT, prophylaxis, prevention, epidemiology, COVID, cancer

## Abstract

Venous thromboembolism (VTE) is the third most common cause of vascular mortality worldwide and comprises deep-vein thrombosis (DVT) and pulmonary embolism (PE). In this review, we discuss how an understanding of VTE epidemiology and the results of thromboprophylaxis trials have shaped the current approach to VTE prevention. We will discuss modern thromboprophylaxis as it pertains to genetic risk factors, exogenous hormonal therapies, pregnancy, surgery, medical hospitalization, cancer, and what is known thus far about VTE in COVID-19 infection.

## 1. Introduction

Venous thromboembolism (VTE) is the third most common cause of vascular mortality worldwide and comprises deep-vein thrombosis (DVT) and pulmonary embolism (PE) [1]. In clinical practice, about two-thirds of VTE episodes manifest as DVT and one-third as PE with or without DVT [2,3]. Less frequently, thrombosis affects other veins including upper extremity veins, cerebral venous sinuses, and mesenteric, renal, and hepatic veins. Venous thromboembolism at uncommon sites is discussed in another article of this VTE compendium [4]. For this review, the term DVT will refer to deep-vein thrombosis involving the veins of the lower extremities, including the inferior vena cava. When VTE, DVT, and PE are discussed these terms refer to symptomatic events except when stated otherwise.

Risk factors for VTE can be subdivided into factors that promote venous stasis, factors that promote blood hypercoagulability, and factors causing endothelial injury or inflammation. These three broad categories, frequently taught as “Virchow’s triad”, have formed the basis for understanding and categorizing the risk factors of VTE for over a century. A clear understanding of the risk factors for VTE is vital to identify patients at risk of VTE who would benefit from thromboprophylaxis. An individual patient’s risk of VTE depends on intrinsic, patient-specific factors (such as genetic risk factors, age, or body mass index) and acquired risk due to the unique context or situation (such as hospitalization, surgery, or cancer). Risk factors are also frequently categorized by “transient vs. persistent” and “major vs. minor”. These distinctions are useful for determining the ongoing risk of VTE and the optimal duration of anticoagulant prophylaxis or treatment.

The primary goal of pharmacologic VTE prophylaxis is to prevent fatal PE, and in the intensive care unit (ICU) and surgical patients pharmacologic VTE prophylaxis is associated with a reduction in mortality [5,6]. Early clinical studies in VTE prevention focused on the potential benefit of prophylactic anticoagulation in populations at high risk of VTE. This approach is easy to implement in patient care, and pharmacologic VTE prophylaxis for medical and surgical inpatients is frequently included in admission order sets for these patients. However, recent evidence suggests the risk of bleeding with anticoagulant thromboprophylaxis negated its benefit in some groups of patients [7,8,9]. Subsequent trials are examining whether it is possible to improve upon this approach by developing a risk prediction tool to identify individuals with high VTE risk who would benefit the most from thromboprophylaxis. In this review, we discuss how VTE epidemiology and the results of contemporary thromboprophylaxis trials have improved our understanding of risk factors and our ability to enhance the benefits of VTE prophylaxis. In particular, we will discuss modern thromboprophylaxis as it pertains to genetic risk factors, exogenous hormonal therapies, pregnancy, surgery, medical hospitalization, cancer, and what is known so far about VTE in COVID-19 infection.

### 1.1. Venous Thromboembolism Remains a Significant Public Health Concern

Pulmonary embolism has been called “the leading cause of preventable death in hospitalized patients” and “the number one priority for improving patient safety in hospitals” [10]. The incidence of VTE is higher in high-income countries than low-income countries, even after adjusting for age, ethnicity, incident cancer diagnoses, body mass index (BMI), and anti-thrombotic therapies [11]. In Western countries, the incidence rates of VTE and PE are approximately 0.87–1.82 per 1000 person-years and 0.45–0.95 per 1000 person-years, respectively [2,11,12,13,14,15,16]. In the United States, this translates to 300,000–600,000 cases of VTE per year, and annual incident VTE events are estimated to cost US healthcare more than 7 billion dollars each year [17]. Rates of VTE are lowest in Asian populations and highest in Northern European populations [18,19]. The patient characteristics and risk factors for VTE also vary by region. A greater proportion of VTE events in Caucasian populations are idiopathic compared to Asian, African, and Hispanic populations [20]. In one study from South Africa, HIV was the most common identifiable risk factor for VTE, and yet in many studies of VTE, data on HIV are not reported [21,22]. Venous thromboembolism and its associated morbidity, mortality, and financial burdens including hospitalization and treatment represent a substantial public health concern.

Assessment of the impact of VTE on our health systems must also consider the burden of patients presenting to the emergency department with suspected VTE, as well as the burden of misdiagnosis. In the general population, only one out of every five patients who are evaluated for VTE in the emergency department for VTE is diagnosed with VTE [23,24]. In pregnant patients, only one in twenty-five patients evaluated for VTE is diagnosed with VTE [23,25]. Consequently, when diagnostic tests are applied to a population with low to moderate VTE incidence, the positive predictive value is moderate (~60%) and false positives are common. Because ruling out VTE often requires clinical evaluation, laboratory testing, and imaging studies, the true healthcare burden of VTE must consider these costs as well as the cost of over- and under-diagnosis [26].

Despite substantial morbidity and mortality, public awareness of VTE remains poor. In a survey of six developed nations, only 28% of respondents were aware of the symptoms of PE and only 19% were aware of the symptoms of DVT [1]. It is thought that increased recognition of the symptoms of DVT and PE could lead to earlier presentation and avoid fatal PE. Initiatives such as “World Thrombosis Day” from the International Society on Thrombosis and Haemostasis (ISTH) seek to advance public awareness of VTE.

### 1.2. Venous Thromboembolism Exists in a Hierarchy of Clinical Importance

Venous thrombi consist predominantly of fibrin and red blood cells [27]. Strong risk factors for VTE such as surgery, trauma, and immobility activate procoagulant proteins and initiate a highly regulated feedback loop which ultimately generates and organizes the fibrin strands that characterize venous thrombosis. Anticoagulant medications used in the prevention of venous thrombosis target critical procoagulant proteins along this pathway to inhibit their activity and prevent the formation of fibrin clots. Most venous thrombi originate in the valve pocket of calf veins where venous stasis is commonly secondary to effects of gravity, venous capacitance, and impaired flow around venous valves. From there, venous clots can propagate proximally along the deep veins of the legs. As clots extend into larger and more proximal vessels, the risk of embolization increases. Clots that embolize travel through successively larger veins returning blood to the heart and into the pulmonary vasculature. From there, emboli enter successively smaller pulmonary arteries and impede circulation when the vasculature becomes too small for the clot to pass.

Not all clots are created equal. Though they share a common pathogenesis, the prognosis of VTE events diverges significantly, and more proximal events carry a higher risk of embolization and potential for mortality [28,29,30]. Venous thromboembolism occurs in a hierarchy of clinical importance, starting with fatal PE, symptomatic PE, symptomatic proximal DVT, asymptomatic proximal DVT, symptomatic distal DVT, and asymptomatic distal DVT. Studies of the epidemiology of VTE often combine these outcomes into a single composite outcome despite their disparate clinical relevance.

Isolated calf DVT is significantly less likely to cause PE, and in many cases, its clinical relevance is uncertain [29,30]. Most distal DVT resolves spontaneously due to the activity of the fibrinolytic system, and distal DVT is a self-limited condition in most patients without ongoing risk factors. In many of the original randomized controlled trials exploring the benefits of pharmacologic prophylaxis with unfractionated heparin (UFH) or low-molecular-weight heparin (LMWH), the primary outcome was driven by the diagnosis of asymptomatic calf DVT by protocol-mandated venography. Asymptomatic DVT by mandated venography is diagnosed 5–21 times more commonly than symptomatic VTE [31], and a meta-analysis of prophylaxis trials in surgical patients found the ratio of asymptomatic to symptomatic thromboses varied widely from 3:1 to 104:1 with a median ratio 14:1 [32]. This wide variation in the ratio of asymptomatic to symptomatic calf vein thrombi confounds attempts to compare thromboprophylaxis strategies. By including these events, studies increase power to show statistically significant outcomes but overestimate the efficacy of antithrombotic agents given by including events of uncertain clinical relevance.

The distinction between proximal and distal DVT is also critical to determining the risk of PE and recurrent DVT after stopping anticoagulation treatment. Conventional wisdom says that isolated distal DVT rarely causes symptomatic PE. Distal DVT events also have a lower risk of recurrence when compared to proximal events. In the absence of identifiable risk factors, the difference in recurrent events may be profound, with recurrence after distal unprovoked DVT occurring with an adjusted hazard ratio of 0.11 (95% CI 0.03–0.45) compared to proximal events [33].

No randomized trials have been conducted that specifically target asymptomatic and incidental VTE, and the optimal approach to treatment of asymptomatic or incidental VTE is extrapolated from studies of patients with symptomatic events [34]. Asymptomatic or incidental PE is detected in approximately 1% of chest CT scans undertaken for indications other than PE [35]. The American College of Chest Physician (ACCP) guidelines advise that patients with asymptomatic VTE should receive the same treatments as those with comparable symptomatic VTE [34]. One important exception to this recommendation is the case of low-risk isolated subsegmental PE with normal ultrasonography of the legs, which should be managed with clinical surveillance over anticoagulation [36].

Over time, the incidence of VTE has increased, but VTE mortality has not [37]. While some of this may be due to advances in treatment, some authors attribute this seemingly incongruous shift to the detection of smaller VTE with lesser clinical relevance. Modern computed tomography scanning techniques can now detect small PE in subsegmental pulmonary arteries that are of uncertain clinical significance. No randomized controlled trials to date have adequately explored the question of whether anticoagulant treatment of isolated subsegmental PE confers clinical benefit [38]. Ongoing trials will explore whether anticoagulation treatments for isolated subsegmental pulmonary emboli meaningfully impact patient-centered outcomes [39]. Even in the presence of malignancy, where treatment of incidental and distal events is generally accepted practice, a lack of evidence precludes definitive recommendations for treatment of subsegmental PE [40].

### 1.3. An Ounce of Prevention is Worth a Pound of Cure

Preventing fatal PE is the primary goal of anticoagulant prophylaxis for VTE. The one-month case fatality rate for VTE ranges from 2.8 to 12% [14,20,41,42]. The case fatality rate from PE accounts for virtually all of the overall case fatality for VTE, and the initial presentation for 24% of PE patients is sudden death [43].

Prevention of VTE also avoids significant post-VTE morbidity. Three conditions that can develop despite appropriate treatment of VTE are post-thrombotic syndrome (PTS), chronic thromboembolic pulmonary hypertension (CTEPH), and post-PE syndrome. Post-thrombotic syndrome is a clinical diagnosis and is composed of a variety of lower extremity symptoms that persist after treatment of DVT including pain, paresthesias, skin pigmentation, functional restriction, and rarely venous stasis ulcers. The incidence of PTS after VTE varies from ~20 to 60% (depending on the definition of PTS) at 1–2 years after VTE diagnosis, with 4.0–5.6% of patients developing debilitating symptoms indicating severe PTS [44,45]. The prevalence of CTEPH after VTE varies significantly in the literature and is estimated at 3–4% [46,47]. The development of CTEPH after PE is not affected by the intensity of anticoagulation after an index PE event and likely represents a pathobiology distinct from acute PE [48]. Long-term impairments to gas exchange and right ventricular dysfunction are the subject of ongoing study under the label “Post-PE syndrome”. Post-PE syndrome encompasses several long-term functional deficits that can occur after acute PE and are associated with a reduced health-related quality of life [49,50]. The prevalence and potential severity of these conditions must be considered when determining the potential benefits of preventing VTE. Averting sudden death and reducing post-PE morbidity are not the only benefits of anticoagulant prophylaxis, and prevention of VTE is important to avoid patient discomfort, anticoagulant treatments and their associated risks, specialist visits, delays in procedures, and the potential for additional testing.

### 1.4. Approaches to Prevent Venous Thromboembolism

There are two major ways to reduce the risk of VTE. The first is to screen patients pre- and post-operatively with accurate diagnostic testing. By diagnosing VTE early, treatment could be provided to halt progression and avoid morbidity and mortality associated with acute VTE. Unfortunately, contrast venography is expensive, painful, and impractical to perform outside of clinical studies [26]. Less invasive studies, such as venous ultrasonography, are less sensitive in asymptomatic patients than in symptomatic patients [51,52]. This is likely because most thrombi are small, non-occlusive calf vein thrombi, most of which may not extend and cause symptomatic DVT or PE and are of uncertain clinical significance. Screening “at-risk” patients is impractical and too expensive to be undertaken outside of clinical trials.

The second approach is to undertake measures to prevent VTE. General measures, such as encouraging early ambulation after surgery, can be adopted universally without harm. In addition, active prophylaxis with either mechanical or pharmacologic means has been proven to lower the risk of VTE. Mechanical prophylaxis refers to devices, such as graduated compression stockings and intermittent pneumatic compression devices, which decrease venous stasis in the lower extremities. Mechanical prophylaxis does not carry a risk of bleeding but can be uncomfortable, and prolonged use can lead to skin breakdown and other cutaneous complications. Recent American Society of Hematology (ASH) guidelines for prophylaxis in medical patients recommend mechanical prophylaxis when the bleeding risk is unacceptably high but suggest using pharmacologic prophylaxis in all patients without elevated risks for bleeding [9]. Given the relative paucity of evidence for mechanical prophylaxis, this review will focus on the use of pharmacologic prophylaxis. Low-dose anticoagulant medications reduce the risk of VTE by 50–80% across a variety of clinical circumstances at a cost of a modest increase in the risk of bleeding. Trials in VTE prevention attempt to identify large groups of patients who are at elevated risk, such as patients after hip surgeries, and administer low dose anticoagulants with the goal of reducing VTE.

Initial studies evaluated “mini dose unfractionated heparin (UFH)” (UFH in doses of 5000 IU every 8–12 h) and warfarin either in low doses or doses that prolonged the international normalized ratio (INR) to 2.0–3.0. These early studies, which were performed in the 1970s and 1980s, used surrogate outcomes such as radioactive fibrinogen uptake leg scanning plus venous ultrasound or contrast venography. These tests were performed and detected asymptomatic calf and proximal vein DVT, assuming that these were reasonable surrogates for symptomatic DVT and PE as well as fatal PE, the latter being clinically relevant. Recent thinking questions this assumption [32,52].

Contemporary studies evaluated parenteral low molecular weight heparins (LMWHs) and direct oral anticoagulants (DOACs) for prevention of DVT and PE and some mandated screening with venography or venous ultrasound. Subsequent consensus guidelines have been developed and published following these studies. Identification of those at highest risk of VTE allows for targeted efforts at prevention. At present, all anticoagulant based strategies of venous thromboprophylaxis increase the risk of bleeding. Consequently, use of both LMWH and low-dose direct oral anticoagulants for prevention of VTE is predicated on a favorable balance of risk and benefit, and the term “net clinical benefit” has been adopted by recent studies evaluating the use of anticoagulant prophylaxis. Averting sudden death and post-VTE morbidity are not the only goals of anticoagulant prophylaxis, and prevention of VTE is also important to avoid patient discomfort, anticoagulant treatments and their associated risks, specialist visits, delays in procedures, and the potential for additional testing.

### 1.5. Age Is One of The Most Important Risk Factors

Even in the presence of cancer, venous access devices, and medications that increase the risk of thrombosis, the incidence of venous thrombosis in children remains low until adulthood—late teens and early 20s. In a study of Korean hospitals, which included 3611 children with cancer over a 15 year period, only 0.9% developed VTE [53]. The vast majority of children diagnosed with VTE have multiple risk factors for thromboembolism, and less than 5% of children with VTE have no provoking factor identified [54]. The incidence of PE is lower in children, but when it is diagnosed, mortality remains substantial at 8–10% [54,55]. The incidence of VTE remains low until patients enter their third decade of life. In contrast to pediatric patients, no significant provoking factor is identified in up to 45–50% of cases of new VTE in adults [14,56].

Up to 60% of all VTE events occur in those over 70 years of age [14]. Several factors may contribute to this increased risk. Procoagulant factors such as factor VIII, factor VII, homocysteine, and fibrinogen increase naturally with age [57]. The development of co-morbidities such as cancer and chronic inflammatory conditions also contribute to the high incidence of VTE in elderly populations.

Age exerts variable effects on the risk of VTE by sex. During childbearing years, the incidence of VTE increases in women, and in the third decade of life, the risk of first VTE events in women exceeds that in men [15,42]. This effect is due to increased endogenous estrogen as well as the increased risk from introduction of exogenous hormonal therapies and pregnancy. Outside of childbearing years, the incidence of VTE is greater in men [14,15].

The risk of VTE increases exponentially with age, and with each decade of life, the absolute risk of VTE increases substantially. The incidence of DVT per 1000 person-years increases from 0.08 for those age 25–29, to 0.39 for age 35–39, 0.82 for age 45–49, 0.91 for age 55–59, 1.13 for age 65–69, 2.94 for age 75–79, and 4.73 for age ≥ 85 [14]. Scoring systems that attempt to define the risk of VTE in individual patients almost always include age as a factor, and physicians should consider the patient’s age as a strong determinant of the risk of VTE when considering VTE prophylaxis.

### 1.6. Genetic Risk Factors

Genetic risk factors vary widely in both their prevalence and their impact on the risk of VTE. The factor V Leiden gene mutation and prothrombin gene mutation are associated with a 3–5-fold increase in the risk of a first episode of VTE, in contrast to antithrombin deficiency, which may increase the risk by 14-fold (see Table 1) [58,59,60]. The most common thrombophilia mutation in Caucasian populations is heterozygous factor V Leiden, occurring in approximately 5% of northern European descendants [61]. In young women with no additional risk factors, the absolute risk of VTE due to heterozygous factor V Leiden is extremely low. The prevalence of prothrombin gene mutation ranges from 0 to 4.7% of asymptomatic individuals and demonstrates significant variation by region [62]. Deficiencies of protein C and S are found in <1% of asymptomatic individuals [63,64].

In the absence of additional acquired risk factors, the presence of a single genetic risk factor is insufficient to recommend long-term anticoagulants. In asymptomatic patients with heterozygous factor V Leiden, heterozygous prothrombin gene mutation, protein C deficiency, and protein S deficiency, the absolute risk of VTE each year is approximately equal to the yearly risk of major bleeding on anticoagulation, and prophylaxis is not recommended for routine use when these mutations are discovered due to a family history of thrombosis [60].

Gene mutations in methylenetetrahydrofolate (MTHFR) and elevated levels of homocysteine and factor VIII have been studied as potential markers of underlying thrombophilia [67,68,69]. Attempts to associate these markers with an underlying predisposition to VTE have not clearly demonstrated their utility in clinical practice. Persistently elevated factor VIII levels (>150% of normal) correlate with the risk of a first episode of VTE, but issues with sample collection, intra-individual variation, and factor levels can lead to difficulty interpreting factor VIII levels [69]. Due to a lack of evidence that these tests can be used to guide effective management, we do not recommend testing for these defects.

One special case which deserves consideration is the presence of antithrombin deficiency, which is a potent inherited risk factor for VTE (OR: 14.0; 95% CI 5.5 to 29.0) and increases the absolute risk of a first VTE to >1% per year [65]. The risk of recurrent VTE for these patients is substantial, and antithrombin deficient patients should be continued on long-term anticoagulation after a first episode of VTE. The optimal management of asymptomatic patients with antithrombin deficiency is not well defined, and consideration for thromboprophylaxis should take into consideration family history and other patient factors that influence the risk of VTE and bleeding. Antithrombin deficiency is an area of ongoing study, but firm conclusions are limited by the small number of patients. Patients with suspected antithrombin deficiency should be evaluated and managed by a thrombosis specialist.

The ability to predict recurrent VTE is not substantially improved by thrombophilia testing, and VTE prophylaxis is not undertaken based on the detection of factor V Leiden, prothrombin gene mutation, or protein C or S deficiencies. The duration of anticoagulation therapy for patients with VTE should be determined on clinical grounds with attention paid to the circumstances and provoking factors of the inciting event rather than underlying genetic factors [70,71]. Small numbers of patients with homozygous factor V Leiden and compound heterozygous factor V Leiden and prothrombin gene mutation preclude definitive conclusions on the risk of recurrence. Conflicting reports exist in the literature as to whether these conditions lead to increased recurrence and are an indication for long-term anticoagulation [72,73,74]. Additionally, identification of thrombophilia does not appear to affect the efficacy of traditional anticoagulant agents, except for antithrombin deficiency, which has the potential to interfere with the action of UFH or LMWH. Identifying asymptomatic genetic risk via thrombophilia testing only rarely affects clinical practice, and genetic studies are of clinical value primarily due to their strong interaction with certain acquired risk factors including estrogen therapy and pregnancy.

## 2. Prevention: Acquired Risk

Over the past three decades, structured protocols have been developed and validated to reliably risk-stratify patients based upon clinical features and laboratory testing into risk groups for VTE. Protocols for estimating the VTE risk of individual patients admitted to hospital, after surgery, and outpatients with cancer are used in clinical practice today to aid in prescribing anticoagulant prophylaxis to those at greatest risk. By combining patient characteristics (e.g., personal history of VTE, BMI, genetic risk factors), context (e.g., hip arthroplasty, ambulatory pancreatic cancer, pregnancy), and predictive biomarkers (e.g., D-dimer) modern scoring systems aim to predict the risk of VTE in individual patients in an effort to target preventative measures to those who will benefit most.

The following sections will focus on patients in specific clinical circumstances and explore the risk of VTE associated with exogenous hormonal therapies, pregnancy, surgery, medical admissions, COVID-19 infections, and cancer. In each of the following sections, we will briefly review the specific burden of VTE, detail any unique interactions with traditional risk factors, outline the current standard approach to VTE prophylaxis if one exists, and address any relevant scoring systems for individualizing VTE risk stratification.

### 2.1. Exogenous Hormonal Therapies

Systemic contraceptive therapies such as estrogen-containing combined oral contraceptive pills (COCs), Nuva-Ring, and Depo Provera injections increase the risk of VTE [75,76,77,78]. The absolute risk is acceptably small in the absence of additional risk factors for VTE, and the benefits to women using these methods can be substantial in avoiding unwanted pregnancy, regulating menstrual cycles, and decreasing bleeding, and a number of other systemic effects. Thrombosis specialists are often asked to evaluate patients with a personal history of VTE, family history of VTE, or asymptomatic thrombophilia mutation to discuss the risks of VTE in these populations. In this section, we will briefly review the data on risk of VTE in patients on exogenous hormonal formulations and then discuss how the absolute risk of VTE changes based on exposure to these treatments.

Over the last sixty years, numerous combined estrogen and progesterone contraceptive pill formulations have been developed. Broadly speaking the use of COCs increases the risk of VTE of 4–6-fold compared to no COCs [77,79]. Estrogen concentration in these combined oral contraceptive pills ranges from 20 to 50 μg, and higher doses of estrogen are associated with an increased risk of VTE. The progestin component of COCs also appears to affect the rate of VTE, with Levonorgestrel and other second-generation progestins having a lower overall VTE risk than formulations with third-generation progestins such as Gesodestrel or Gestodene (see Table 2) [77,80]. The risk of VTE is highest in the first year after starting COC or hormone replacement therapy, likely representing an “unmasking” period wherein patients who are predisposed to VTE develop an event after developing an additional risk factor [81,82,83]. The risk also increases in the presence of additional risk factors including advancing age, and those on COCs after age 35 are at increased risk of VTE.

Less is known about the VTE risk in individuals on exogenous testosterone therapy. Testosterone supplementation may increase the risk of venous thromboembolic disease, and ongoing use is correlated with increased VTE across a wide age range and without respect to endogenous testosterone levels [86].

To understand and counsel patients on the risk of VTE associated with hormonal therapies or genetic risk factors, we advise discussing risk in the form of absolute risk of VTE per year. For this example, we assume women with increased VTE risk are appropriately prescribed lower-risk COCs that use levonorgestrel and lower doses of estrogen. The incidence of VTE in women of child-bearing age who are not taking oral contraceptives is approximately 1 in 10,000 per year. The baseline risk of VTE for exogenous estrogen exposure through use of the combined oral contraceptive pill increases the risk of VTE by approximately 4-fold to 4 in 10,000 per year. In those with a family history of VTE, the risk doubles to 8 in 10,000 per year even in the absence of identifiable thrombophilia. The presence of heterozygous factor V Leiden and the oral contraceptive pill have a multiplicative effect, increasing the risk of thrombosis to 35–40/10,000 per year, generously approximated at 0.5% risk of VTE per year [58,76]. In this situation, patients should be offered alternative options for contraception that do not increase the risk of VTE, but each of these options has varying risks and benefits and may not be suitable for all patients. As an example, a patient with heterozygous factor V Leiden initiating oral contraception may choose to accept a 0.5% per year risk of VTE, and these patients should be educated on the signs and symptoms of VTE and to seek medical attention urgently if any of these symptoms should occur.

After VTE associated with exogenous estrogen exposure or pregnancy, the risk of recurrent VTE is significantly elevated in any future situation where estrogen exposure is increased. This includes the future use of COCs, hormone replacement therapy, and any future pregnancies. One option for VTE prophylaxis in patients who choose to continue COCs or hormone replacement therapy after a first VTE event is to maintain anticoagulation therapy throughout the period of exogenous estrogen exposure. For many women, this is undesirable with respect to both cost and an increased risk of bleeding. Options for contraception after estrogen-associated VTE that do not increase the risk of VTE include barrier methods, progestin-only contraceptive pills, and both copper and progesterone-based intra-uterine devices [88,89].

### 2.2. Pregnancy

Pregnancy is an independent risk factor for VTE, and VTE is responsible for approximately 10% of maternal mortality in advanced health systems [90]. Pregnancy causes venous stasis because of pelvic venous compression by the gravid uterus, and compression of the left iliac vein by the right iliac artery, leading to sluggish flow in the left iliac vein and its tributaries. Due to differential compressive effects, DVTs in the left leg greatly outnumber DVTs in the right leg during pregnancy [91]. In addition, the risk of thrombosis may be elevated due to increased endogenous procoagulant activity and reduced anticoagulant activity that occur in pregnancy to limit blood loss during childbirth. Endogenous thrombin potential increases over all three trimesters, peaking at the time of delivery. The risk of thrombosis peaks at 1–3 weeks postpartum and remains elevated until up to 12 weeks postpartum [90]. Low-molecular-weight heparin is preferable to unfractionated heparin for prophylaxis in pregnancy to decrease the risks of heparin-associated osteoporosis and heparin-induced thrombocytopenia [92].

It has been recommended that anticoagulant prophylaxis be offered to pregnant women with a greater than 2% risk of VTE in the 9-month antepartum period and a greater than 1% risk of VTE in the 6-week postpartum period [93]. This “risk threshold” was determined by a panel of experts, considering the cost, discomfort, and potential harms of subcutaneous anticoagulant prophylaxis including an increased risk of major bleeding by up to 1% for those receiving pharmacologic prophylaxis [92,93,94]. Conditions and historical factors that cause the risk of thrombosis to exceed these thresholds are detailed in the ASH 2018 Guidelines on thrombosis in pregnancy [93]. Patients with a prior history of VTE are at risk of VTE in pregnancy and warrant consideration of thromboprophylaxis. The optimal management of patients with incidental detection of thrombophilia is less clear, and recommendations differ depending on the underlying genetic condition identified.

A few genetic conditions confer a particularly high risk of VTE in pregnancy and warrant special consideration. Those at highest risk include pregnant patients with antithrombin deficiency, homozygous factor V Leiden, and compound heterozygotes with both factor V Leiden and prothrombin gene mutations. In the setting of antithrombin deficiency without pharmacologic VTE prophylaxis, available risk estimates range from 6.1 to 11.6% across the antepartum and postpartum period [95,96]. In one small retrospective study, the risk of VTE remained high despite LMWH prophylaxis, with 7.0% (95% CI 1.8–17.8) of patients experiencing a VTE event [95]. The patients in that study did not receive antithrombin substitution, and expert consultation with consideration for antithrombin concentrates has been suggested in high-risk patients to further reduce the risk of VTE [97,98]. In patients over 35 years of age with homozygous factor V Leiden, the risk of VTE during pregnancy approximates 3.4%, and in compound heterozygotes for factor V Leiden and prothrombin gene mutation, the risk was 8.2% [96].

Approaches to individualized risk stratification such as clinical scoring systems have not been well validated in the pregnant population. Prior scoring systems have relied heavily on either personal history of VTE or genetic risk factors, and ultimately these other approaches closely approximate the results of less complex approaches that rely on these factors in isolation [99,100,101,102]. In cases where the benefits of pharmacologic prophylaxis are not clear, additional risk factors that could be considered include maternal age > 35, BMI > 30 kg/m^2^, twin pregnancy, and immobility as included in the Lyon VTE score [100], or stillbirth, pre-term birth, prolonged labor, infection, hemorrhage, and blood transfusion as noted by the most recent Royal College of Obstetricians and Gynaecologists (RCOG) guidelines [101,103]. In the future, studies exploring early risk stratification of pregnant patients may allow us to consider primary prophylaxis for those patients at the highest risk of VTE, just as they have in surgical patients, hospitalized populations, and ambulatory cancer patients.

### 2.3. Surgery

Up to 20% of all new cases of VTE have recent surgery as a provoking factor [14]. Increased procoagulant activity, reduced fibrinolysis, venous stasis, and reduced mobility all contribute to increased risk. After major surgery, the risk of VTE is substantially elevated for at least 12 weeks. This risk is concentrated in the first six weeks, where studies show a relative risk of 69.1 (95% CI 63.1–75.6) compared to no surgery [104].

The absolute risk of VTE after orthopedic surgery varies depending on the population studied. In a Northern European population expected to be high risk for VTE, the cumulative incidence of VTE was 0.73% within 30 days and 0.83% within 31–365 days after hip fracture surgery [105]. Prior to hospital discharge, VTE occurs in 1.09% of total hip arthroplasty patients and 0.53% of total knee arthroplasty patients [106]. In Korean patients, the risk of PE after total hip and knee replacement approximates 0.44% [107]. When VTE is diagnosed after total hip arthroplasty, mortality is estimated at 30% across a variety of populations, with an absolute risk of fatal PE of approximately 0.2% [105,106,107].

A variety of anticoagulant agents have been studied for use in preventing post-operative VTE. The use of pharmacologic treatment for preventing VTE in this setting reduces the risk of symptomatic and fatal VTE by up to 70% [5]. The duration of treatment and specific agents used are dependent on the type of surgery performed. Only 35% of post-operative VTE occurs while patients are still in hospital, and measures to prevent thrombosis must include preventative treatments prescribed in the outpatient setting [108].

In contemporary trials, with appropriate prophylaxis, rates of VTE are reduced to approximately 1.0%, and fatal PE rates are reduced to 0.3% [32]. With current prophylaxis, early ambulation, and improved operative techniques, rates of thrombosis may be decreasing further [109]. Minimally invasive abdominal surgeries have become more common over the past decade. Approaches with laparoscopy have a lower rate of venous thromboembolic disease compared with laparotomy in patients undergoing gynecologic procedures [110]. Early ambulation after surgery is believed to be responsible for a reduction in VTE risk, but the effect is difficult to measure.

Current approaches to pharmacologic VTE prophylaxis in surgery favor fixed-duration prophylaxis where the type of surgery informs the duration of VTE prophylaxis. Surgeries which substantially increase the risk of VTE include abdominal surgery for cancer and total hip replacement. For these procedures, extended prophylaxis is likely appropriate for a duration of up to 30–35 days based on the elevated VTE risk during this period [111]. For many other procedures such as total knee replacement, the risk of VTE is lower, and 14 days of prophylaxis is likely appropriate.

For individualized VTE prophylaxis after surgery, the most widely used tool is the CAPRINI score. This scoring system has been validated across multiple surgical subtypes including critically ill surgical patients and is able to risk-stratify patients for post-operative VTE [112,113,114,115,116,117]. Risk assessment models require patient-specific information to generate an estimate of that individual’s risk of VTE (see Figure 1). The CAPRINI score is best used with an online calculator and estimates an individual’s risk of VTE over the first three months after surgery. This estimate can provide clinicians with guidance on which patients are at highest risk, and who may benefit from initial and extended pharmacologic prophylaxis after surgery.

### 2.4. Medical Hospitalization

Most hospitalizations are for non-surgical conditions. Between 1 and 3% of patients admitted to hospital will suffer a complication of VTE during hospitalization, and many have increased risk after hospital discharge [118,119]. Risk factors for VTE in hospitalized patients include both patient-specific factors and acquired disease states. A meta-analysis demonstrated that a history of VTE was the strongest predictive factor in the development of VTE. Older age, elevated CRP, D-dimer, fibrinogen levels, heart rate, thrombocytosis, leukocytosis, fever, leg edema, immobility, paresis, previous history of VTE, thrombophilia, malignancy, critical illness, and infections are all risk factors for the development of VTE in hospitalized patients [120]. The absolute risk of VTE in individual medical patients is low, but the total burden of VTE from hospitalized medical patients is substantial. Estimates indicate that 25–30% of all VTE in the US is related to medical hospitalization. This represents approximately 150,000 patients per year in the US [121]. In one estimate, VTE related to hospitalizations accounted for 71% of all VTE related death [41].

Anticoagulant prophylaxis to prevent VTE results in a 50–70% reduction in the rate of VTE [122,123]. Results from randomized controlled trials for anticoagulant prophylaxis of medical inpatients are largely driven by detection of asymptomatic, and distal DVT by mandated venography or compression ultrasound. Whether VTE prophylaxis in hospitalized patients reduces mortality remains a contentious issue, and in studies, the benefits of VTE reduction in these patients are often offset by the risk of bleeding. In one large registry in the US, the use of pharmacologic VTE prophylaxis was associated with improved mortality in critically ill patients compared to no prophylaxis or mechanical VTE prophylaxis alone [6].

In the absence of contraindications, the administration of low-molecular-weight heparin, unfractionated heparin, or fondaparinux to at-risk medical inpatients to prevent VTE has become standard-of-care across most of the world. Current guidelines recommend routine in-hospital thromboprophylaxis for at-risk medical patients [9]. Scoring systems, including the Padua Predictive Score, the International Medical Prevention Registry on Venous Thromboembolism score (IMPROVE VTE), and IMPROVE-DD scores assist with individualizing VTE prophylaxis in medical patients [119,124,125,126,127,128]. These scoring systems take into account variables known to increase the risk of VTE including a prior history of VTE, immobility, malignancy, stroke, advanced age, body mass index, acute infection, and in the case of the IMPROVE-DD score, the use of a biomarker—D-dimer greater than 2 times the upper limit of normal—to risk stratify patients [119,124,126,129]. The NHS in England and the Center for Medicare and Medicaid Services in the US have mandated the use of standardized VTE Risk Assessment Models (RAMs) to guide the use of thromboprophylaxis for inpatients. Computer alert interventions and other systemic measures to ensure appropriate VTE assessment are effective at optimizing the implementation of VTE prophylaxis and reducing preventable VTE [130,131]. A recent policy statement from the American Heart Association discusses the benefits of these interventions and provides recommendations for improving VTE prevention in hospitalized patients [132]. Due to heterogeneity in the VTE risk of medical inpatients, risk assessment and provision of thromboprophylaxis to those at higher VTE risk increases the net clinical benefit from thromboprophylaxis and avoids unnecessary provision of thromboprophylaxis to patients at low risk.

As in surgical populations, 45–57% of VTE related to medical hospitalization occur during the hospitalization itself [118,119]. The risk of VTE is highest immediately after hospitalization and remains elevated for up to 3 months after admission. Recently, two studies examined the benefit of extending thromboprophylaxis with oral options but did not provide definite evidence of net clinical benefit [133,134]. Routine use of extended thromboprophylaxis after discharge without individualized risk assessment is not recommended [9]. A recent publication in this VTE series explores the recent trials for extended VTE prophylaxis and suggests an algorithm to determine which patients may benefit from extended prophylaxis [135].

### 2.5. COVID-19 Infection

Immune activation to fight bacterial, viral, and fungal pathogens stimulates a complex cascade of inflammatory cytokines [136]. This in turn leads to activation of thrombin and vascular endothelial injury, increasing the risk of venous thromboembolic disease. This activation exists on a spectrum from beneficial to pathologic, and in severe inflammatory conditions, the risk of VTE can increase dramatically. During ICU admissions, the need for prolonged immobilization, indwelling central venous catheters, and hypoxia due to acute respiratory distress syndrome are additional risk factors for development of thrombosis, which in part explains the high rate of VTE seen in patients with severe COVID-19 infection [137].

Several mechanisms contribute to the unusually high rate of venous thrombosis seen in COVID-19 patients. Infection of pulmonary tissues via angiotensin-converting enzyme 2 (ACE2) receptors leads to direct endothelial injury, and the immune response leads to the release of pro-thrombotic cytokines [138]. Thromboelastographic studies have demonstrated a hypercoagulable state and decreased fibrinolysis in COVID-19 patients [139,140,141]. In addition to the conventional mechanism for PE, a preponderance of evidence suggests that immunothrombosis contributes to in situ development of pulmonary artery thrombi [138,142,143]. In addition to these disease-specific factors, patients admitted with suspected or confirmed COVID-19 are hypoxic and may be asked to limit their ambulation for the purpose of infection control, both of which are potent risk factors for VTE.

Data on VTE in asymptomatic, mild, and ambulatory COVID-19 patients are limited, and the risk of thrombosis in these settings is not well established. Early reports have shown the risk of VTE in hospitalized and severe COVID-19 infection is increased compared to patients with similar severity of illness who do not have COVID-19. One series of COVID-19 patients in hospital but outside the ICU showed VTE incidence of 3.8%, an incidence of PE 2.5%, and a rate of positive imaging tests of 46%, potentially indicating underdiagnosis in this population [144]. By comparison, a study of hospitalized COVID-19 patients screened asymptomatic patients with D-dimer > 1000 ng/mL with whole leg compression ultrasound and found asymptomatic DVT in 14.7% of patients, but nearly all of these asymptomatic events were distal DVT [145].

Rates of VTE in COVID-19 patients requiring ICU care are even higher. An early report from an ICU in Wuhan utilized screening ultrasound to show DVT occurred in 25% (20/81) of patients admitted with COVID-19 pneumonia [146]. Notably, the baseline risk of VTE is lower in China, and thromboprophylaxis is not routinely used in these patients. Recent data reveal that increased rates of VTE are seen even with the use of prophylaxis. An ICU series in Italy reported 22% of patients developed VTE [147]. A publication from France compared the frequency of PE in 107 COVID-19 patients with a control period and with ICU admissions for influenza. In COVID-19 patients, the frequency of pulmonary embolism was 20.6% compared to 6.1% in the control group and 7.5% in the group of patients with influenza [148]. In a study of two French intensive care units that utilized systematic screening with compression ultrasound 1–3 days after admission, the proportion of patients with VTE was 69%, with 23% of patients diagnosed with pulmonary embolism despite prophylaxis, and in some cases despite therapeutic anticoagulation [149,150]. In a series of hospitalized patients in the Netherlands with only 37% of patients admitted to ICU, the incidence of VTE was 11% (95% CI 5.8–17%) and 23% (95% CI 14–33%) at 7 days and 14 days, respectively [151]. With the addition of screening, the overall VTE was 34% at 14 days despite weight-adjusted VTE prophylaxis with Nadroparin [151]. Follow-up data confirm a very high incidence of PE in this population—approximately 25% [142].

The development of VTE in COVID-19 patients appears to correlate with more severe disease course, higher D-dimer, higher CRP, and mortality [142,146,151]. High D-dimer > 1.0 μg/mL and a Padua prediction score ≥ 4 may be useful markers for predicting DVT in hospitalized COVID-19 patients and require further study [152].

There are limitations and challenges with interpreting these early reports because of their mainly retrospective designs, short duration of follow-up, and consequent concerns about bias including publication and case ascertainment biases. Nonetheless, with a rate of VTE > 20% in populations of severe COVID-19, many physicians are calling for increased doses of prophylactic anticoagulation or the empiric use of therapeutic dose anticoagulation [141,151,153,154]. However, intensifying anticoagulant therapy may increase the risk of bleeding, particularly in those who are critically ill. Ongoing randomized trials will determine whether these approaches will improve patient outcomes.

### 2.6. Cancer-Associated Thrombosis

In patients with cancer, the risk of VTE is substantially increased due largely to tumor production of procoagulant and inflammatory substances which are released into the circulation. Rates of VTE in malignancy approximate 1% per year with significant variation based on tumor subtype from 0.5 to 20% per year [155]. Interactions between chemotherapy anticancer treatments and the development of VTE are also well established. Gemcitabine and Cisplatin chemotherapy are strongly associated with VTE risk [156,157]. All solid malignancies appear to increase the risk of VTE, and certain cancer sites including pancreatic and gastric cancer have particularly high rates of VTE during initial treatment. Patients with hematologic malignancies are also at increased risk of VTE, except for some indolent lymphoma subtypes [158].

For pancreatic ductal adenocarcinoma, the incidence of VTE approaches 20% over the first twelve months after diagnosis, and this number may be even higher in populations with higher proportions of metastatic disease [159,160]. On autopsy, more than 40% of pancreatic cancer patients are affected by VTE [161]. Patients with cancer-associated thrombosis typically have a poor prognosis, and in these patients, VTE likely serves as an indicator for more aggressive disease. Similar observations over many years have led to attempts to risk-stratify and prevent thrombosis in cancer patients.

The Khorana score is the most widely known tool by which ambulatory cancer patients are risk-stratified using clinical and laboratory criteria. Points are assigned for a pre-chemotherapy platelet count ≥ 350, hemoglobin level < 10 g/dL (or RBC growth factors), pre-chemotherapy leukocyte count > 11 × 10⁹/L, and BMI ≥ 35 kg/m². Pancreatic cancer and stomach cancer each count for two points, and lymphoma, gynecologic, bladder, and testicular cancer count for one point. In the original retrospective and prospective validation cohorts, patients with a score ≥ 3 had a 7.1% and 6.7% risk of VTE at a median of 2.5 months, respectively [162]. Multiple prospective and series have confirmed the validity of the Khorana score, and its use is endorsed by the most recent ASCO guidelines for VTE risk stratification in cancer [40,163,164]. The Khorana score is also valid for use in hospitalized cancer patients [165,166]. Attempts to validate the Khorana score for specific subpopulations of cancer have been less successful, and in studies of lung cancer, hepatocellular carcinoma, acute myeloid leukemia, and lymphoid malignancies, the Khorana score did not adequately stratify or predict VTE events [167,168,169,170].

The recent CASSINI and AVERT studies used this risk stratification model to identify high-risk patients who would benefit from DOAC thromboprophylaxis. These trials randomized patients with ambulatory cancers initiating chemotherapy with Khorana score ≥ 2 to placebo or prophylactic rivaroxaban or apixaban, respectively. The AVERT trial showed a significant 6% (4.2% vs. 10.2%, HR 0.41, 95% CI 0.26–0.65, *p* < 0.001) absolute decrease in the rate of symptomatic and incidental VTE and an absolute increase in ISTH major bleeding of 1.7% (3.5% vs. 1.8%, HR 2.00, 95% CI 1.01–3.95, *p* = 0.046), but no cases of fatal bleeding were observed [171]. Major bleeding events are more frequent in cancer patients, due to both the propensity of certain cancer subtypes to bleeding complications (including gastrointestinal and genitourinary cancers) and the tendency of cancer patients towards anemia and blood transfusion.

The CASSINI trial also enrolled patients with a Khorana score ≥ 2 but excluded those with primary or metastatic brain cancer. The composite primary endpoint of DVT, PE, and VTE-related death occurred in 5.95% of patients in the rivaroxaban group and 8.79% in the placebo group (HR 0.66, 95% CI 0.40–1.09, *p* = 0.10). The rates of major bleeding and fatal bleeding were also not statistically different between the study arms (HR 1.96, 95% CI 0.59–6.49, *p* = 0.265) with one fatal bleeding event observed in the rivaroxaban group [172]. Both CASSINI and AVERT were analyzed over a period of six months, but the mortality rates and drug discontinuation rates were much higher in CASSINI than AVERT, both of which may be attributable to more severe underlying disease in the CASSINI patients (mortality approximately 20% CASSINI vs. 12% AVERT, drug discontinuation rate 47% CASSINI vs. 18% AVERT). Of the 62 patients who had a primary end-point event, 24 (39%) did so after stopping the study drug. A pre-specified intervention-period analysis in CASSINI, which considers only time on the drug, demonstrated a statistically significant reduction in the rate of VTE (2.6% vs. 6.4% HR 0.40 95% CI 0.20–0.80). Although the CASSINI trial did not meet its primary efficacy endpoint, the findings are consistent with those of AVERT. The high discontinuation rates of the DOACs over time in both trials and in treatment trials highlight the challenges with adherence to DOACs in the prevention and treatment of VTE in patients with cancer [173,174,175].

By leveraging our understanding of VTE risk to target patients at the highest risk of VTE, these studies have attempted to maximize the net clinical benefit of preventative treatments for VTE. Similarly, patients at relatively low risk of VTE are spared the expense and potential bleeding complications of these treatments. When targeted appropriately to these high-risk groups, oral VTE prophylaxis with DOACs is cost-effective [176], and a meta-analysis comparing the efficacy of DOAC to LMWH for VTE prevention in cancer confirms a similar reduction in the risk of VTE [177].

Attempts to further refine the Khorana score include the Cancer-And-Thrombosis-Study (CATS) score, which combines tumor subtype and D-dimer levels to predict those at greatest risk of cancer-associated thrombosis [178]. Elevated CRP, creatinine, and nodal involvement may also be markers of VTE risk [179]. Further research is required to determine if additional clinical or laboratory parameters can be leveraged to improve our ability to predict the development of VTE and target prophylactic measures accordingly.

## 3. Future Directions for Prevention of Venous Thromboembolism

Advances in technology and pharmacology have already improved our ability to predict and prevent VTE. At least two major barriers to improving VTE prevention still exist. The first is the limitations on overall benefit to thromboprophylaxis inherent to current anticoagulant medications. The second barrier is the imperfect science of individualizing VTE risk stratification and the inherent complexity of predicting multifactorial and competing phenomena. These final sections will discuss future directions in therapeutics and technology including the importance of antithrombotic options with less bleeding and technological advances including machine learning to refine risk stratification and facilitate their implementation.

### 3.1. A More Palatable Approach to Pharmacologic Prevention

Over the past decade, the tolerability, acceptability, and quality of life for patients at risk of VTE have changed with the advent of effective oral therapies for VTE treatment and prevention. In those with cancer, patient-reported quality of life is better with oral anticoagulant treatment compared to daily subcutaneous LMWH injection [180]. After only one month on therapy, those taking subcutaneous injections experienced excess bruising, stress, worry and irritation, and frustration taking anticoagulants compared to those on oral therapy. While not as rigorously studied, quality-of-life improvements with oral therapies are likely similar in the prophylactic setting.

Increased tolerability and ease of use has led to more widespread adoption and adherence to VTE prophylaxis in populations such as ambulatory cancer patients and post-operative patients. Oral options for prophylaxis reduce the time required for counseling and reviewing self-injection technique or administering subcutaneous medications by healthcare practitioners and increase patient adherence to anticoagulant medications.

### 3.2. Bleeding Complications Limit the Net Benefits of Thromboprophylaxis

Every effective anticoagulant medication for VTE prophylaxis also increases the risk of bleeding. The harms of bleeding—including substantial case fatality rates—may be under-appreciated. Many authors have argued that declining rates of VTE in hospitalized patients should be cause for re-appraisal and de-escalation of previous protocols for the use of VTE prophylaxis [7,8]. As the baseline rate of VTE changes, either through time or differing circumstance, we must balance the threats of both underuse and overuse of pharmacologic VTE prophylaxis.

Novel agents for pharmacologic prophylaxis are currently being developed that target factor XI and factor XII. Knockout of factor XI and XII in animal models reduces thrombosis without increasing bleeding. Early studies with factor XI antisense oligonucleotides were successful in lowering factor XI levels and preventing VTE without significant effects on hemostasis and bleeding [181]. Whether this result is confirmed with oral or long-acting parenteral factor XI inhibitors in larger studies is a question of significant interest. If these trials or other new agents are successful at reducing VTE without substantially increasing the risk of bleeding, existing approaches to VTE prevention could change dramatically.

### 3.3. Implementing and Improving Scoring Systems

Efforts to predict and prevent venous thromboembolic disease are predicated on our ability to accurately identify patients at risk. The benefits of thromboprophylaxis must be weighed against the financial costs and potential for increased bleeding. Understanding which patients are at greatest risk can help medical practitioners and their patients to make intelligent decisions regarding the use of anticoagulants to prevent VTE. We must continue to collect and analyze large data sets on patient-specific and acquired risk factors, and how they interact, to improve existing risk assessment models. Early successes with predicting VTE risk using biomarkers should prompt further research into the use of biomarkers in new situations. The discovery and evaluation of novel biomarkers to risk-stratify patients may be an area of significant interest going forward, and serum biomarkers such as Vascular Endothelial Growth Factor (VEGF), Interferon-alpha (IFN-α), Interleukin-15 (IL-15), and Citrullinated histone H3 (H3cit) may see further investigation for this purpose [182,183,184].

Biomarkers are only one aspect of prediction. The development and validation of prediction models should seek to increase the accuracy of prediction without sacrificing usability [185,186]. As we seek to individualize preventative efforts, the primary obstacles that affect implementation of risk assessment models will be the complications and complexity of any proposed algorithm. Scoring systems and other risk-assessment models can be implemented for use by health professionals if they are easily accessible and understood, but developing sufficient predictive power for VTE often requires the combination of several variables. Scoring systems can quickly become time-consuming for use by physicians, and hiring additional data entry personnel adds significant cost. Predictive algorithms that appropriately consider and calculate bleeding risks for individual patients to avoid harms from VTE prophylaxis increase complexity and add additional workload. Scoring systems developed in the current age must not lose sight of the practical challenges of implementation by treating clinicians.

### 3.4. Better Prevention Through Technology

Electronic medical records (EMRs) have provided new opportunities for VTE prevention. Scoring systems and other decision support tools can be incorporated into EMRs for ease of access. Studies indicate that computer alert interventions can increase adherence to appropriate VTE risk-stratification [187], reduce costs by avoiding unnecessary thromboprophylaxis in low-risk patients [188], and decrease preventable harm from VTE [131,132,188]. Implementation of such systems should consider whether the absolute benefit is worth the additional burden on healthcare providers as long as direct provider input is required for the system to function.

Smoother implementation of existing categorical scoring systems is laudable, but health technologies in the future will render such systems obsolete. By taking continuous variables such as age or weight and reducing them to binary or ternary variables, we necessarily sacrifice some predictive value. Predicting an individual’s risk of VTE is extremely challenging because no single variable is strongly predictive, and we are forced to rely on systems that incorporate multiple variables to produce meaningful predictive values for VTE. Machine learning has the capacity to overcome these challenges by recognizing patterns in complex sets of information that accurately predict the risk of VTE and bleeding [189]. When interpreted by such systems, continuous variables do not need to be reduced to categorical variables for ease of use. Additionally, new clinical events can be fed into machine learning algorithms continuously, which in turn would allow such systems to adjust the weight of variables in risk calculations quickly and precisely.

Until such technology is available, institutional protocols for VTE prophylaxis should be regularly reviewed to ensure they reflect recent studies detailing risks and benefits of this common medical intervention. Discussions at the institutional and individual level should seek to balance the complex relationship between the multifactorial risk factors for VTE and bleeding complications and strive for net clinical benefit in prescribing pharmacologic VTE prophylaxis.

## 4. Conclusions

In the last half-century, we have made tremendous progress in understanding the epidemiology and prevention of VTE. During that time, we have moved from studies detailing the benefits of “mini-dose” unfractionated heparin given to post-operative patients to sophisticated algorithms that leverage patient-specific and acquired risk factors to determine which patients will derive the greatest benefit from VTE prophylaxis. Patients considering exogenous hormonal therapies or pregnancy can be accurately informed about their risk for VTE, and patients undergoing surgery, medical hospitalization, cancer treatments, and those with COVID-19 infection routinely receive preventative anticoagulant treatments when the benefits of such treatments are known to outweigh the risks. As scoring systems and other decision support tools increase in accuracy and complexity, we risk overwhelming clinicians, or worse, turning them into data entry personnel. In the future, additional biomarkers for predicting VTE, safer medications, and machine learning algorithms will revolutionize prediction and prevention of this common disorder.

## Figures and Tables

**Figure 1 jcm-09-02467-f001:**
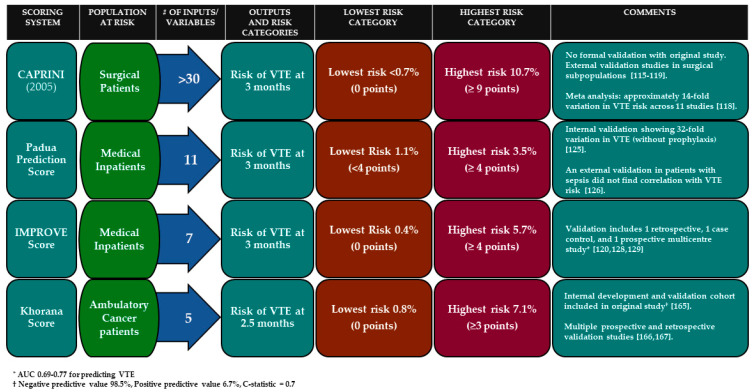
Summary of select scoring systems for predicting risk of venous thromboembolism.

**Table 1 jcm-09-02467-t001:** Association of Patient-Specific Risk Factors With Risk of First Episode of VTE [65,66,67].

Intrinsic Risk Factors	Odds Ratio (or) for VTE
Elevated BMI > 30	2.3
Heterozygous Prothrombin gene mutation	2.8
Heterozygous Factor V Leiden gene mutation	4.2
Homozygous Prothrombin gene mutation	6.7
Homozygous Factor V Leiden gene mutation	11.5
Antithrombin deficiency	14.0

**Table 2 jcm-09-02467-t002:** Association of Acquired Risk Factors With The Risk of First Episode of VTE [42,76,84,85,86,87].

Acquired Risk Factors	Odds Ratio (or) for VTE
Seated immobility at work *	1.8
Long-Haul Travel ^†^	2.1
Testosterone supplementation	2.3
Low risk COC (Levonorgestrel)	3.6
Pregnancy or Postpartum	4.2
Trauma/Fracture	4.6
Medical Hospitalization	5.1
Neurologic Disease with Leg Paresis	6.1
High risk COC (Desogestrel)	6.8
Active Cancer	14.6
Surgery	21.7

* Seated at least eight hours a day and at least three hours at a time without getting up; † Greater than four hours in a seated position in plane, car, bus, or train.

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
