# Peer review of "Prevention of Venous Thromboembolism in 2020 and Beyond"

_jcm, 2020, doi:10.3390/jcm9082467_

Round 1
Reviewer 1 Report
In this Review entitled « Prevention of Venous Thromboembolism in 2020 and Beyond », Nicholson et al provided an overview of the risk factors of venous thromboembolism and emphasized different preventive tools and the future directions for venous thromboembolism (VTE) prevention.
This is a subject of particular interest in regard with the great impact on morbidity and mortality of VTE in hospitalized patients, especially during COVID-19 era. Nicholson et al highlighted the main situations encountered in clinical practice. The different mechanisms which lead to VTE and the tools used to assess VTE risk are well-described.
My major concerns are as follow:
- The authors should add the reference of and discuss the recent AHA policy statement about the venous thromboembolism prevention in hospitalized patients (Henke et al Circulation 2020)
- Section about thromboprophylaxis in COVID-19 is one of the strengths of this review in regard with the current context. However, this part should not be limited to severe forms of COVID-19 since some reports highlighted the strong incidence of venous thrombotic events in non-ICU patients (Zhang et al Circulation, Lodigani et al Thromb Res, Demelo-Rodriguez et al Thromb Res). Reports concerning the effect of thromboprophylaxis on prognosis should be cited (for instance Paranjpe et al JACC). In addition, many references are missing to support cited data.
- I’m not sure that the section “15. Future Directions for VTE Prevention” is necessary in this current form. It could be merged with the sections 16, 17 and 18 in a section “Future directions”.
- While the number of references is already important, some references are missing to support assertions, definitions and epidemiological data (see bellow).
I have several minor comments:
- Lines 19-20: “Venous thromboembolism (VTE) is the third most common cause of vascular mortality worldwide”. Please add reference
- Lines 20-22: “In clinical practice about two-thirds of VTE episodes manifest as DVT and one-third as PE with or without DVT”. Please provide reference
- Lines 43-43: “However, recent evidence suggests the risk of bleeding with anticoagulant thromboprophylaxis negated its benefit in some groups of patients”. Please add reference
- Lines 54-56: there is a problem in references number
- Lines 71-72: “Because ruling out VTE often requires clinical evaluation and imaging studies the true healthcare burden of VTE must consider these costs”. Ruling out VTE is also based on D-Dimers, at least for patients with low probability of VTE. Please add reference of guidelines
- Line 80: “Venous thrombi consist predominantly of fibrin and red blood cells”. Please provide references
- Line 97: “dinicholsonmsparate” is probably a typo error. Please amend.
- Lines 149-157: please add reference to support the discussion on performances and inconveniences of VTE screening.
- Lines 200-201: “Procoagulant factors such as factor VIII, factor VII, homocysteine, and fibrinogen increase naturally with age”. Please provide a reference
- Lines 210-215: please amend to avoid repetition
- Lines 215-217: “The most common thrombophilia mutation in Caucasian populations is heterozygous factor V Leiden occurring in approximately 5% of northern European descendants”. Please add reference
- Lines 228-235: references are missing to support this interesting paragraph
- Lines 458-464: please add references
- Line 466: please define acronym
- Lines 467-468: “Thromboelastographic studies in Italy have also demonstrated decreased fibrinolysis in severe COVID-19 patients”. Please provide the reference
- Lines 477-478: The patients in this report were the first of study center, not of France. Please precise the sentence.
- Line 501: “Gemcitabine and Cisplatin chemotherapy are strongly associated with VTE risk”. Please add reference.
- Lines 562-563: “Early studies with Factor XI antisense oligonucleotides were successful in lowering factor XI levels and preventing VTE without significant effects on hemostasis and bleeding”. Please add reference.
Author Response
Response Letter to Reviewers
Re: jcm-863082
We thank the reviewers for their thoughtful and detailed recommendations and believe their concerns can be satisfactorily addressed by the revised manuscript. In this letter, we describe changes made to address these concerns and provide explanatory text where required. Attached is a copy of the revised manuscript with many of the revisions highlighted.
Regarding the points raised by reviewer 1
In this Review entitled « Prevention of Venous Thromboembolism in 2020 and Beyond », Nicholson et al provided an overview of the risk factors of venous thromboembolism and emphasized different preventive tools and the future directions for venous thromboembolism (VTE) prevention.
This is a subject of particular interest in regard with the great impact on morbidity and mortality of VTE in hospitalized patients, especially during COVID-19 era. Nicholson et al highlighted the main situations encountered in clinical practice. The different mechanisms which lead to VTE and the tools used to assess VTE risk are well-described.
My major concerns are as follow:
The authors should add the reference of and discuss the recent AHA policy statement about the venous thromboembolism prevention in hospitalized patients (Henke et al Circulation 2020)
We agree. Included in the section on hospitalized patients are two new sentences that discuss implementation in more detail including explicit reference to this publication: (lines 473-483)
“The NHS in England and the Center for Medicare and Medicaid Services in the US have mandated the use of standardized VTE Risk Assessment Models (RAMs) to guide the use of thromboprophylaxis for inpatients. Computer alert interventions and other systemic measures to ensure appropriate VTE assessment are effective at optimizing the implementation of VTE prophylaxis [133,134]. A recent policy statement from the American Heart Association discusses the benefits of these interventions and provides recommendations for improving VTE prevention in hospitalized patients [135]. Due to heterogeneity in the VTE risk of medical inpatients, risk assessment and provision of thromboprophylaxis to those at higher VTE risk increases the net clinical benefit from thromboprophylaxis and avoids unnecessary provision of thromboprophylaxis to patients at low risk.”
Additionally, information in Henke publication is relevant to our discussion on the use of technology (680-685) and is cited using the original sources:
“16. Better Prevention Through Technology
Electronic medical records (EMRs) have provided new opportunities for VTE prevention. Scoring systems and other decision support tools can be incorporated into EMRs for ease of access. Studies indicate computer alert interventions can increase adherence to appropriate VTE risk-stratification [165], reduce costs by avoiding unnecessary thromboprophylaxis in low-risk patients [166], and decrease preventable harm from VTE [126,127,166]. “
Section about thromboprophylaxis in COVID-19 is one of the strengths of this review in regard with the current context. However, this part should not be limited to severe forms of COVID-19 since some reports highlighted the strong incidence of venous thrombotic events in non-ICU patients (Zhang et al Circulation, Lodigani et al Thromb Res, Demelo-Rodriguez et al Thromb Res). Reports concerning the effect of thromboprophylaxis on prognosis should be cited (for instance Paranjpe et al JACC). In addition, many references are missing to support cited data
This is an important point and we have added the following text
Lines (509-545)
“Data on VTE in asymptomatic, mild, and ambulatory COVID-19 patients are limited, and the risk of thrombosis in these settings is not well established. Early reports have shown the risk of VTE in hospitalized and severe COVID-19 infection is increased compared to patients with similar severity of illness who do not have COVID- 19. One series of COVID-19 patients in hospital but outside the ICU showed VTE incidence of 3.8%, an incidence of PE 2.5%, and a rate of positive imaging tests of 46% potentially indicating underdiagnosis in this population [148]. By comparison a study of hospitalized COVID-19 patients screened asymptomatic patients with D-dimer >1000ng/mL with whole leg compression ultrasound and found asymptomatic DVT in 14.7% of patients, but nearly all of these asymptomatic events were distal DVT [149].
Rates of VTE in COVID-19 patients requiring ICU care are even higher. An Early report from an ICU in Wuhan utilized screening ultrasound to show DVT occurred in 25% (20/81) of patients admitted with COVID-19 pneumonia [150]. Notably the baseline risk of VTE is lower in China, and thromboprophylaxis is not routinely used in these patients. Recent data reveals that increased rates of VTE are seen even with the use of prophylaxis. An ICU series in Italy reported 22% of patients developed VTE [151]. A publication from France compared the frequency of PE in 107 COVID-19 patients with a control period and with ICU admissions for influenza. In COVID-19 patients the frequency of pulmonary embolism was 20.6% compared to 6.1% in the control group and 7.5% in the group of patients with influenza [152]. In a study of two French intensive care units that utilized systematic screening with compression ultrasound 1-3 days after admission the proportion of patients with VTE was 69%, with 23% of patients diagnosed with pulmonary embolism despite prophylaxis, and in some cases despite therapeutic anticoagulation [153,154]. In a series of hospitalized patients in the Netherlands with only 37% of patients admitted to ICU the incidence of VTE was 11% (95% CI 5.8-17%) and 23% (95% CI 14-33%) at 7 days and 14 days, respectively [155]. With the addition of screening the overall VTE was 34% at 14 days despite weight-adjusted VTE prophylaxis with Nadroparin [155]. Follow-up data confirms a very high incidence of PE in this population – approximately 25% [146].
The development of VTE in COVID-19 patients appears to correlate with more severe disease course, higher D-dimer, higher CRP, and mortality [146,150,155]. High D-dimer >1.0 μg/mL and a Padua prediction score ≥4 may be useful markers for predicting DVT in hospitalized COVID-19 patients and require further study [156].
There are limitations and challenges with interpreting these early reports because of their mainly retrospective designs, short duration of follow-up, and consequent concerns about bias including publication and case ascertainment biases. Nonetheless, with a rate of VTE >20% in populations of severe COVID-19, many physicians are calling for increased doses of prophylactic anticoagulation or the empiric use of therapeutic dose anticoagulation [144,155,157,158]. However, intensifying anticoagulant therapy may increase the risk of bleeding, particularly in those who are critically ill. Ongoing randomized trials will determine whether these approaches will improve patient outcomes.”
- Klok, F.A.; Kruip, M.J.H.A.; van der Meer, N.J.M.; Arbous, M.S.; Gommers, D.; Kant, K.M.; Kaptein, F.H.J.; van Paassen, J.; Stals, M.A.M.; Huisman, M. V.; et al. Confirmation of the high cumulative incidence of thrombotic complications in critically ill ICU patients with COVID-19: An updated analysis. Thromb. Res. 2020, doi:10.1016/j.thromres.2020.04.041.
- Fox, S.E.; Akmatbekov, A.; Harbert, J.L.; Li, G.; Quincy Brown, J.; Vander Heide, R.S. Pulmonary and cardiac pathology in African American patients with COVID-19: an autopsy series from New Orleans. Lancet Respir. Med. 2020, 8, 681–686, doi:10.1016/S2213-2600(20)30243-5.
- Lodigiani, C.; Iapichino, G.; Carenzo, L.; Cecconi, M.; Ferrazzi, P.; Sebastian, T.; Kucher, N.; Studt, J.D.; Sacco, C.; Alexia, B.; et al. Venous and arterial thromboembolic complications in COVID-19 patients admitted to an academic hospital in Milan, Italy. Thromb. Res. 2020, 191, 9–14, doi:10.1016/j.thromres.2020.04.024.
- Demelo-Rodríguez, P.; Cervilla-Muñoz, E.; Ordieres-Ortega, L.; Parra-Virto, A.; Toledano-Macías, M.; Toledo-Samaniego, N.; García-García, A.; García-Fernández-Bravo, I.; Ji, Z.; de-Miguel-Diez, J.; et al. Incidence of asymptomatic deep vein thrombosis in patients with COVID-19 pneumonia and elevated D-dimer levels. Thromb. Res. 2020, 192, 23–26, doi:10.1016/j.thromres.2020.05.018.
- Cui, S.; Chen, S.; Li, X.; Liu, S.; Wang, F. Prevalence of venous thromboembolism in patients with severe novel coronavirus pneumonia. J. Thromb. Haemost. 2020, doi:10.1111/jth.14830.
- Tavazzi, G.; Civardi, L.; Caneva, L.; Mongodi, S.; Mojoli, F. Thrombotic events in SARS-CoV-2 patients: an urgent call for ultrasound screening. Intensive Care Med. 2020, 3–5, doi:10.1007/s00134-020-06040-3.
- Poissy, J.; Goutay, J.; Caplan, M.; Parmentier, E.; Duburcq, T.; Lassalle, F.; Jeanpierre, E.; Rauch, A.; Labreuche, J.; Susen, S. Pulmonary Embolism in COVID-19 Patients: Awareness of an Increased Prevalence. Circulation 2020, 1–6, doi:10.1161/circulationaha.120.047430.
- Llitjos, J.-F.; Leclerc, M.; Chochois, C.; Monsallier, J.-M.; Ramakers, M.; Auvray, M.; Merouani, K. High incidence of venous thromboembolic events in anticoagulated severe COVID-19 patients. J. Thromb. Haemost. 2020, doi:10.1111/jth.14869.
- Helms, J.; Tacquard, C.; Severac, F.; Leonard-Lorant, I.; Ohana, M.; Delabranche, X.; Merdji, H.; Clere-Jehl, R.; Schenck, M.; Fagot Gandet, F.; et al. High risk of thrombosis in patients with severe SARS-CoV-2 infection: a multicenter prospective cohort study. Intensive Care Med. 2020, 46, 1089–1098, doi:10.1007/s00134-020-06062-x.
- Middeldorp, S.; Coppens, M.; van Haaps, T.F.; Foppen, M.; Vlaar, A.P.; Müller, M.C.A.; Bouman, C.C.S.; Beenen, L.F.M.; Kootte, R.S.; Heijmans, J.; et al. Incidence of venous thromboembolism in hospitalized patients with COVID‐19. J. Thromb. Haemost. 2020, jth.14888, doi:10.1111/jth.14888.
- Zhang, L.; Feng, X.; Zhang, D.; Jiang, C.; Mei, H.; Wang, J.; Zhang, C.; Li, H.; Xia, X.; Kong, S.; et al. Deep Vein Thrombosis in Hospitalized Patients with Coronavirus Disease 2019 (COVID-19) in Wuhan, China: Prevalence, Risk Factors, and Outcome. Circulation 2020, 114–128, doi:10.1161/circulationaha.120.046702.
- Thomas, W.; Varley, J.; Johnston, A.; Symington, E.; Robinson, M.; Sheares, K.; Lavinio, A.; Besser, M. Thrombotic complications of patients admitted to intensive care with COVID-19 at a teaching hospital in the United Kingdom. Thromb. Res. 2020, 191, 76–77, doi:10.1016/j.thromres.2020.04.028.
- Paranjpe, I.; Fuster, V.; Lala, A.; Russak, A.J.; Glicksberg, B.S.; Levin, M.A.; Charney, A.W.; Narula, J.; Fayad, Z.A.; Bagiella, E.; et al. Association of Treatment Dose Anticoagulation With In-Hospital Survival Among Hospitalized Patients With COVID-19. J. Am. Coll. Cardiol. 2020, 76, 122–124, doi:10.1016/j.jacc.2020.05.001.
I’m not sure that the section “15. Future Directions for VTE Prevention” is necessary in this current form. It could be merged with the sections 16, 17 and 18 in a section “Future directions”.
Unfortunately, the original major section-headings were lost, and “major” and “minor” section headings were merged before they were distributed to reviewers. The paragraph in 15. is intended to represent brief introductory remarks at the outset of a new major section. We have re-inserted the major headings: I., II. III. Major section III has been reworked to improve the balance between the subsections within it. Below is the text, including a new line to clarify what is intended and signpost the following subsections.
(Lines 614-622, formerly section 15)
“III. Future Directions for Prevention of Venous Thromboembolism
Advances in technology and pharmacology have already improved our ability to predict and prevent VTE. At least two major barriers to improving VTE prevention still exist. The first are the limitations on overall benefit to thromboprophylaxis inherent to current anticoagulant medications. The second barrier is the imperfect science of individualizing VTE risk, and the inherent complexity of predicting multifactorial and competing phenomena. These final sections will discuss future directions in therapeutics and technology including the importance of antithrombotic options with less bleeding, and advances in technology such as machine learning to refine risk stratification and to facilitate the implementation of risk stratification strategies. “
While the number of references is already important, some references are missing to support assertions, definitions and epidemiological data (see below).
Thank you. We have added new citations to support each of the point described below.
I have several minor comments:
Lines 19-20: “Venous thromboembolism (VTE) is the third most common cause of vascular mortality worldwide”. Please add reference
References added: Wendelboe 2016
Lines 20-22: “In clinical practice about two-thirds of VTE episodes manifest as DVT and one-third as PE with or without DVT”. Please provide reference
References added: Anderson 1991; Oger 2000
Lines 43-43: “However, recent evidence suggests the risk of bleeding with anticoagulant thromboprophylaxis negated its benefit in some groups of patients”. Please add reference
References added: Millar 2015; Novo-Veleiro 2018; Schünemann 2018
Lines 54-56: there is a problem in references number
Inserted reference using reference manager appropriately (Siegal 2020) [11] – subsequent numbering should now be accurate throughout.
Lines 71-72: “Because ruling out VTE often requires clinical evaluation and imaging studies the true healthcare burden of VTE must consider these costs”. Ruling out VTE is also based on D-Dimers, at least for patients with low probability of VTE. Please add reference of guidelines
Added explicit reference to laboratory testing separate from clinical evaluation in this line. A separate article in this review series will address diagnosis of VTE in detail including current guidelines. The text was also updated to reflect concerns from another reviewer regarding over and under diagnosis of PE. We now cite the 2012 Diagnosis of DVT: Antithrombotic therapy and prevention of thrombosis, 9th ed: American College of Chest Physicians evidence-based clinical practice guidelines [27]
Revised text: (line 68-76)
Assessment of the impact of VTE on our health systems must also consider the burden of patients presenting to the emergency department with suspected VTE, as well as the burden of misdiagnosis. In the general population only one out of every five patients who are evaluated for VTE in the emergency department for VTE are diagnosed with VTE [24,25]. In pregnant patients only one in twenty-five patients evaluated for VTE are diagnosed with VTE [24,26]. Consequently, when diagnostic tests are applied to a population with low to moderate VTE incidence, the positive predictive value is moderate (~60%) and false positives are common. Because ruling out VTE often requires clinical evaluation, laboratory testing, and imaging studies the true healthcare burden of VTE must consider these costs as well as the cost of over and under diagnosis [27].
Line 80: “Venous thrombi consist predominantly of fibrin and red blood cells”. Please provide references
Reference added: Walton 2015
Line 97: “dinicholsonmsparate” is probably a typo error. Please amend.
Corrected.
Lines 149-157: please add reference to support the discussion on performances and inconveniences of VTE screening
References were added to the paragraph below (now lines 164-172)
“There are two major ways to reduce the risk of VTE. The first is to screen patients pre- and post-operatively with accurate diagnostic testing. By diagnosing VTE early, treatment could be provided to halt progression and avoid morbidity and mortality associated with acute VTE. Unfortunately contrast venography is expensive, painful, and impractical to perform outside of clinical studies [27]. Less invasive studies, such as venous ultrasonography, are less sensitive in asymptomatic patients than in symptomatic patients [51,52]. This is likely because most thrombi are small, non-occlusive calf vein thrombi, most of which may not extend and cause symptomatic DVT or PE and are of uncertain clinical significance. Screening ‘at risk’ patients is impractical and too expensive to be undertaken outside of clinical trials.”
Some information in the paragraph stems from the collective experience and opinions of the authors.
- Bates, S.M.; Jaeschke, R.; Stevens, S.M.; Goodacre, S.; Wells, P.S.; Stevenson, M.D.; Kearon, C.; Schunemann, H.J.; Crowther, M.; Pauker, S.G.; et al. Diagnosis of DVT: Antithrombotic therapy and prevention of thrombosis, 9th ed: American College of Chest Physicians evidence-based clinical practice guidelines. Chest 2012, 141, e351S-e418S, doi:10.1378/chest.11-2299.
- Johnson, S.A.; Stevens, S.M.; Woller, S.C.; Lake, E.; Donadini, M.; Cheng, J.; Labarère, J.; Douketis, J.D. Risk of deep vein thrombosis following a single negative whole-leg compression ultrasound: a systematic review and meta-analysis. JAMA 2010, 303, 438–45, doi:10.1001/jama.2010.43.
- Hirsh, J.; Ginsberg, J.S.; Chan, N.; Guyatt, G.; Eikelboom, J.W. Mandatory contrast-enhanced venography to detect deep-vein thrombosis (DVT) in studies of DVT prophylaxis: Upsides and downsides. Thromb. Haemost. 2013, 111, 10–13, doi:10.1160/TH13-07-0562.
Lines 200-201: “Procoagulant factors such as factor VIII, factor VII, homocysteine, and fibrinogen increase naturally with age”. Please provide a reference
Now lines 219-220 - References added: Mari 2008
Lines 210-215: please amend to avoid repetition
We have written a more concise version of this section that also now references the table. (lines 235-239)
“Genetic risk factors vary widely in both their prevalence and impact on the risk of VTE. The factor V Leiden gene mutation and prothrombin gene mutation are associated with a 3-5 fold in the risk of a first episode of VTE in contrast to Antithrombin deficiency which may increase the risk by 14-fold (See Table 1) [59–61]. The most common thrombophilia mutation in Caucasian populations is heterozygous factor V Leiden occurring in approximately 5% of northern European descendants [62].”
Lines 215-217: “The most common thrombophilia mutation in Caucasian populations is heterozygous factor V Leiden occurring in approximately 5% of northern European descendants”. Please add reference
Reference added: Rees 1995 as above [62]
Lines 228-235: references are missing to support this interesting paragraph
References added: Hirmerová 2013; Schambeck 2001; Simone 2013;
Lines 458-464: please add references
Reference to inflammatory cytokines and coagulation in infection: Levi 2003
Reference added to specific risk factors: Minet 2015
The revised paragraph (Now 493-499)
Immune activation to fight bacterial, viral, and fungal pathogens stimulates a complex cascade of inflammatory cytokines [139]. This in turn leads to activation of thrombin and vascular endothelial injury, increasing the risk of venous thromboembolic disease. This activation exists on a spectrum from beneficial to pathologic, and in severe inflammatory conditions the risk of VTE can increase dramatically. During ICU admissions the need for prolonged immobilization, indwelling central venous catheters, and hypoxia due to ARDS are additional risk factors for development of thrombosis, which in part explains the high rate of VTE seen in patients with severe COVID-19 infection [140].
Line 466: please define acronym
Revised as follows (now line 502)
Infection of pulmonary tissues via angiotensin-converting enzyme 2 (ACE2) receptors leads to direct endothelial injury and the release of pro-thrombotic cytokines [141].
Lines 467-468: “Thromboelastographic studies in Italy have also demonstrated decreased fibrinolysis in severe COVID-19 patients”. Please provide the reference
Removed reference to Italy.
Revised sentence now reads:
Thromboelastographic studies have demonstrated a hypercoagulable state and decreased fibrinolysis in COVID-19 patients [142–144]
References: Wright 2020; Panigada 2020; Maatman 2020;
Lines 477-478: The patients in this report were the first of study center, not of France. Please precise the sentence.
Removed reference to first patients. Revised text (now 520-524)
A publication from France compared rates of PE in 107 COVID-19 patients with a control period and with ICU admissions for influenza. In COVID-19 patients the rate of pulmonary embolism was 20.6% compared to 6.1% in the control group and 7.5% in the group of patients with influenza [152]
Line 501: “Gemcitabine and Cisplatin chemotherapy are strongly associated with VTE risk”. Please add reference.
References added: Seng 2012; Rupa-Matysek 2018;
Lines 562-563: “Early studies with Factor XI antisense oligonucleotides were successful in lowering factor XI levels and preventing VTE without significant effects on hemostasis and bleeding”. Please add reference
Reference added
Büller, H.R.; Bethune, C.; Bhanot, S.; Gailani, D.; Monia, B.P.; Raskob, G.E.; Segers, A.; Verhamme, P.; Weitz, J.I. Factor XI antisense oligonucleotide for prevention of venous thrombosis. N. Engl. J. Med. 2015, 372, 232–240, doi:10.1056/NEJMoa1405760.
We thank the reviewer for their comments.

Reviewer 2 Report
The authors provide an extensive and very well written review.
Nevertheless there are few aspects, that might have the potential to improve the paper:
line 95: (hierarchy of clinical importance) The issue of "asymptomatic or incidental PE" (excessively discussed not only in cancer pts) is missing. In fact it seems important to me, to differentiate between symptomatic (sE) and asymptomatic (asE) events throughout the text (at best) or at least to define what is meant by "PE" and "VTE". Are these always sE ?
For example: the statement on PE and distal DVT (line 113) is correct only when excluding asPE (small vene==> small thrombi==>non-symptomatic PE; the clinical role of repetitive asPE is open)!
Furthermore the extention of a distal DVT to become proximal cannot be excluded by a single "initial" diagnostic procedure and thus distal DVT may not be "clinically irrelevant" (line 156).
Another important issue is the inconsistent "grade" of recommendations. Some are (correctly) very concrete (e.g. line 404), others refer to citations (e.g. 456). A more homogeneous systematic recommendation (weak to strong ??) may be helpful for clinicians.
Less text and a few more tables may be helpful to summarize systems for sVTE risk assessment.
Another (missing) issue for VTE prevention in the ambulatory and post-discharge setting is compliance or persistence, perfectly shown in prevention and treatment trials of cancer-associated VTE (CAT).
For CAT-studies the ISTH-system of bleeding upgrades minor bleeding event to "major" due to the fact that cancer pts (w or w/o anticancer treatment) have a greater tendency to anemia or the need for blood transfusion. Thus an increase in "major ISTH bleeding" in cancer pts does not mean the same as in non-cancer pts. Letal bleeding seems not to be a major matter of concern in anticoagulation for prevention (and probably not for treatment).
line 435: please add: ...heparin and fondaparinux
Some of the abbrevations are not explained (HRT, THA,TKA).
Table 1 relates to first sVTE. (VTE recurrence risk is different)
Author Response
Response Letter to Reviewers
Re: jcm-863082
We thank the reviewers for their thoughtful and detailed recommendations and believe their concerns can be satisfactorily addressed by the revised manuscript. In this letter, we describe changes made to address these concerns and provide explanatory text where required. Attached is a copy of the revised manuscript with revisions highlighted.
Regarding the points raised by reviewer 2:
The authors provide an extensive and very well written review.
Nevertheless there are few aspects, that might have the potential to improve the paper:
line 95: (hierarchy of clinical importance) The issue of "asymptomatic or incidental PE" (excessively discussed not only in cancer pts) is missing. In fact it seems important to me, to differentiate between symptomatic (sE) and asymptomatic (asE) events throughout the text (at best) or at least to define what is meant by "PE" and "VTE". Are these always sE ?
For example: the statement on PE and distal DVT (line 113) is correct only when excluding asPE (small vene==> small thrombi==>non-symptomatic PE; the clinical role of repetitive asPE is open)!
These points are well taken. Several parts of the opening sections have been edited to clarify this point and minimize repetition in the remainder of the manuscsript. Highlighted text is new.
(Line 26-27)
“When VTE, DVT, and PE are discussed these refer to symptomatic events except when stated otherwise.”
Lines 115-120 contain the most important information on symptomatic vs. asymptomatic VTE as it relates to thromboprophylaxis trials. This paragraph now transitions into a paragraph with additional information on asymptomatic VTE in lines 121-129
(115-120)
“Asymptomatic DVT by mandated venography are diagnosed 5-21 times common than symptomatic VTE [32], and a meta-analysis of prophylaxis trials in surgical patients found the ratio of asymptomatic to symptomatic thromboses varied widely from 3:1 to 104:1 with a median ratio 14:1 [33]. This wide variation in the ratio of asymptomatic to symptomatic calf vein thrombi confounds attempts to compare thromboprophylaxis strategies. By including these events studies increase power to show statistically significant outcomes but overestimate the efficacy of antithrombotic agents given by including events of uncertain clinical relevance.
(121-129)
No randomized trials have been conducted that specifically target asymptomatic and incidental VTE, and the optimal approach to treatment of asymptomatic or incidental VTE is extrapolated from studies of patients with symptomatic events [34]. Asymptomatic or incidental PE is detected in approximately 1% of chest CT scans undertaken for indications other than PE [35]. The American College of Chest Physician (ACCP) guidelines advise that patients with asymptomatic VTE should be receive the same treatments as those with comparable symptomatic VTE [34].”
Wording was updated in line 117 (formerly line 113) for clarity:
“Conventional wisdom says that isolated distal DVT rarely causes symptomatic PE.”
In this case we include the word symptomatic because much of the discussion in the last few paragraphs is about the distinction between aVTE and sVTE. After this section references to symptomatic DVT have been removed, and the reader is to assume sVTE is the default VTE as stated in the opening paragraph.
Remaining exceptions to this rule are few including
“By comparison a study of hospitalized COVID-19 patients screened asymptomatic patients with D-dimer >1000ng/mL with whole leg compression ultrasound and found asymptomatic DVT in 14.7% of patients, but nearly all of these asymptomatic events were distal DVT [144].” (lines 514-517)
“The AVERT trial showed a significant 6% (4.2% vs 10.2%, HR 0.41, 95% CI 0.26-0.65, p < 0.001) absolute decrease in the rate of symptomatic and incidental VTE and an absolute increase in ISTH major bleeding of 1.7%...” (lines 581)
Which require delineation of asymptomatic and incidental events due to the nature of the studies being quoted.
Furthermore the extension of a distal DVT to become proximal cannot be excluded by a single "initial" diagnostic procedure and thus distal DVT may not be "clinically irrelevant" (line 156).
Agree – poor choice of words in this case. Edited text is below (now line 170-171)
“This is likely because most thrombi are small, non-occlusive calf vein thrombi, most of which do not extend and cause symptomatic DVT or PE and are of uncertain clinical significance.”
Another important issue is the inconsistent "grade" of recommendations. Some are (correctly) very concrete (e.g. line 404), others refer to citations (e.g. 456). A more homogeneous systematic recommendation (weak to strong ??) may be helpful for clinicians.
As you noted in some cases recommendations are stronger than others. Where sufficient evidence exists and particularly where guidelines give strong recommendations (e.g. 404) we present these as definitive statements. Conversely in some areas where strong evidence is lacking (e.g. 456, 351-359, etc.) we refer to expert opinion by citing their work. If there any specific examples where you are concerned about the wording, we are happy to make changes, but after reviewing the text I am optimistic our readers can distinguish the strength of these recommendations appropriately.
Less text and a few more tables may be helpful to summarize systems for sVTE risk assessment.
Comparing systems for risk assessment is a challenging task due to the heterogeneous nature of their development, validation, and reporting. Figure 1 (below) has been developed to provide a high-level summary of scoring systems that are in use today and appears after the first mention of a scoring system in the figure
(Line 442 - apologies but please see attached in the manuscript as the file won't load in this response system)
|
Another (missing) issue for VTE prevention in the ambulatory and post-discharge setting is compliance or persistence, perfectly shown in prevention and treatment trials of cancer-associated VTE (CAT).
There has already been discussion amongst the co-authors about whether to include information from CASSINI or the meta-analysis on DOACs in cancer - it is difficult to introduce these without a fulsome discussion due to the complexities of interpreting a “negative” study. We agree with your suggestion and now discuss the issue of adherence and persistence (Lines 599-601) highlighted at the bottom of the paragraph. Additionally, new paragraphs discussing the salient points of CASSINI including the important point you raise regarding the importance of adherence. The relevant updated text is as follows (576-601)
“The recent CASSINI and AVERT studies take this risk stratification model to identify high risk patients who would benefit from DOAC thromboprophylaxis. These trials randomized patients with ambulatory cancers initiating chemotherapy with Khorana score ≥ 2 to placebo or prophylactic rivaroxaban or apixaban and respectively. The AVERT trial showed a significant 6% (4.2% vs 10.2%, HR 0.41, 95% CI 0.26-0.65, p < 0.001) absolute decrease in the rate of symptomatic and incidental VTE and an absolute increase in ISTH major bleeding of 1.7% (3.5% vs 1.8%, HR 2.00, 95% CI 1.01-3.95, p = 0.046) but no cases of fatal bleeding were observed [175]. Major bleeding events are more frequent in cancer patients, both due to the propensity of certain cancer subtypes to bleeding complications (including gastrointestinal and genitourinary cancers) and the tendency of cancer patients towards anemia and blood transfusion.
The CASSINI trial also enrolled patients with a Khorana score ≥ 2 but excluded those with primary or metastatic brain cancer. The composite primary end point of DVT, PE, and VTE-related death occurred in 5.95% of patients in the rivaroxaban group and 8.79% in the placebo group (HR 0.66, 95% CI 0.40-1.09, p = 0.10). The rates of major bleeding and fatal bleeding were also not statistically different between the study arms (HR 1.96, 95% CI 0.59-6.49, p = 0.265) with one fatal bleeding event observed in the rivaroxaban group [176]. Both CASSINI and AVERT were analyzed over a period of six months, but the mortality rates and drug discontinuation rates were much higher in CASSINI than AVERT, both of which may be attributable to more severe underlying disease in the CASSINI patients (Mortality approximately 20% CASSINI vs. 12% AVERT, drug discontinuation rate 47% CASSINI vs. 18% AVERT). Of the 62 patients who had a primary end-point event, 24 (39%) did so after stopping the study drug. A pre-specified intervention-period analysis in CASSINI, which considers only time on the drug, demonstrated a statistically significant reduction in the rate of VTE (2.6% vs. 6.4% HR 0.40 95% CI 0.20-0.80). Although the CASSINI trial did not meet its primary efficacy endpoint, the findings are consistent with those of AVERT. The high discontinuation rates of the DOACs over time in both trials and in treatment trials highlight the challenges with adherence to DOACs in the prevention and treatment of VTE in patients with cancer [177-179].”
For CAT-studies the ISTH-system of bleeding upgrades minor bleeding event to "major" due to the fact that cancer pts (w or w/o anticancer treatment) have a greater tendency to anemia or the need for blood transfusion. Thus an increase in "major ISTH bleeding" in cancer pts does not mean the same as in non-cancer pts. Letal bleeding seems not to be a major matter of concern in anticoagulation for prevention (and probably not for treatment).
The above paragraph which now includes the rates of fatal bleeding for both the AVERT and CASSINI trials (0 and 1 respectively). We also highlight this important point in the following addition:
(582-585) are highlighted in the above text where we address this important point.
“Major bleeding events are more frequent in cancer patients, both due to the propensity of certain cancer subtypes to bleeding complications (including gastrointestinal and genitourinary cancers) and the tendency of cancer patients towards anemia and blood transfusion.”
line 435: please add: ...heparin and fondaparinux
Thank you. See the revised text below. (Now line 464-467)
“In the absence of contraindications, the administration of low-molecular-weight heparin, unfractionated heparin, or fondaparinux to at- risk medical inpatients to prevent VTE has become standard-of-care across most of the world. Current guidelines recommend routine in-hospital thromboprophylaxis for at-risk medical patients [9]”
Some of the abbrevations are not explained (HRT, THA,TKA)
All three abbreviations have been removed from the text and replaced with their expanded form. In the sentence below the word “symptomatic” has also been removed prior to the first use of the word VTE to avoid repetition.
(Lines 407-411)
“Prior to hospital discharge VTE occurs in 1.09% of total hip arthroplasty patients and 0.53% of total knee arthroplasty patients [109]. In Korean patients the risk of PE after total hip and knee replacement approximates 0.44% [110]. When VTE is diagnosed after total hip arthroplasty mortality is estimated at 30% across a variety of populations, with an absolute risk of fatal PE of approximately 0.2% [108-110].”
Table 1 relates to first sVTE. (VTE recurrence risk is different)
Table titles have been amended for precision of language – see below.
Table 1. The Association of Patient-Specific Risk Factors With Risk of First Episode of VTE [57–59].
|
INTRINSIC RISK FACTORS |
ODDS RATIO (OR) FOR VTE |
|
Elevated BMI >30 |
2.3 |
|
Heterozygous Prothrombin gene mutation |
2.8 |
|
Heterozygous Factor V Leiden gene mutation |
4.2 |
|
Homozygous Prothrombin Gene mutation |
6.7 |
|
Homozygous Factor V Leiden gene mutation |
11.5 |
|
Antithrombin Deficiency |
14.0 |
Table 2. Association of Acquired Risk Factors With The Risk of First Episode of VTE [73–78].
|
ACQUIRED RISK FACTORS |
ODDS RATIO (OR) FOR VTE |
|
Seated immobility at work* |
1.8 |
|
Long-Haul Travel† |
2.1 |
|
Testosterone supplementation |
2.3 |
|
Low risk COC (Levonogestrel) |
3.6 |
|
Pregnancy or Postpartum |
4.2 |
|
Trauma/Fracture |
4.6 |
|
Medical Hospitalization |
5.1 |
|
Neurologic Disease with Leg Paresis |
6.1 |
|
High risk COC (Desogestrel) |
6.8 |
|
Active Cancer |
14.6 |
|
Surgery |
21.7 |
We thank the reviewer for their comments.

Reviewer 3 Report
The review "Prevention of Venous Thromboembolism in 2020 and Beyond" by Nicholson et al is a comprehensive work of VTE associated risk factors and prophylaxis.
The review is well-written and reads well. However, there are things to be discussed.
General considerations:
- Given the fact that the manuscript is described as a review nothing is mentioned on how the literature was searched or evaluated. The review could definitely benefit from some more information on the methods used.
- As a reader the text could benefit from some reorganization of the headings. Before section 6, a heading stating "Risk factors for VTE" could be relevant. Also, I am confused about section 8. The text continues describing risk factors so why this middle section?
- The tables uses odds ratios. However, it is easier to interpret a HR or RR. Also, often the OR are not referred to in the text making the tables less useful. Not all risk factors are mentioned in the table, e.g. COVID19. The headings of the tables: could consider using the word 'association' instead of 'effect'.
- Line 67-72: Could be relevant to discuss positive predictive values of diverse diagnostic tests and additionally cost/benefits of using the tests.
- Line 95-98 and line 131: could discuss under-diagnosing of fatal PE. Historical autopsy studies all describe large proportion of un-recognised PE ante-mortem.
- Section 6: should be mentioned that the risk differentiate according to sex. At a younger age, women have the highest risk, later men exceed.
- Section 7: nothing is mentioned on thrombophilia testing. Please deliberate this.
- Please be consistent on using either 'VTE' or 'venous thromboembolism'.
- It is not consistent if the headings uses the term 'VTE' or if only the risk factor is mentioned (e.g. section 9 on hormonal therapy). I suggest not mentioning VTE in headings.
- Consider including a table on risk scores and associated purpose. Could also mention existing risk scores for VTE recurrence risk (DOI: 10.1055/s-0040-1708877)
- When describing a new risk factor, please begin with describing the size of the problem. Section 13 line 464: when referring to a 'high rate' but nothing is described yet on these rates. Similar for other sections (e.g. section 14).
- Please be aware, that a proportion is not similar to a rate. A rate must always contain a time measure as opposed to a proportion (percentage) that is without a time scale. Example: line 480: proportions referred to as rates.
- Prediction models: when describing prediction models and mentioning external validation please refer to performance measures of the models. Especially the calibration would be relevant in the VTE setting.
- Section 14: noting is mentioned on DOACs for cancer-associated VTE?
- Section 15 is too short. More could be said about future directions. It seems section 15 to 18 are not very well-balanced. More nuance is needed to strengthen the text in the end.
- The text could benefit from a conclusion in the end to summarize the findings of the review.
Minor comments:
- please limit the use of abbreviations COC/OCP (not explained in table), HRT (line 292), ICU (line 462), ARDS (line 463).
- Line 37 and line 129: I agree that the worst consequence is fatal PE. However, it could be mentioned that preventing all types of VTE is important when using prophylaxis.
- Additional references are needed: line 94 (after 'mortality'), line 100 (after 'uncertain'), line 123 (the sentence starting with 'ongoing'), line 277 (after 'VTE'), 342 (after 'period'), line 375 (after 'surgery'), line 453 (after 'benefit').
- Space is needed: line 39, line 540.
- Line 56: BMI should be abbreviated first time used
- Line 76 "earlier presentation" is not similar to preventing VTE. Line 149: 'reduce the risk' not the same as 'prevent VTE'. Line 150: 'diagnosing early VTE' not the same as 'prevent VTE'.
- line 135: please clarify that PTS is a clinical diagnosis.
- Line 158: unsure what is meant by 'undertake active measures to prevent VTE'.'
- line 160: it would be relevant here to mention the uncertainties regarding duration of OAC and findings of extended treatment in medically ill, e.g. studies on Betrixaban.
- Section 6: should be mentioned that the risk differentiate according to sex. At a younger age, women have the highest risk, later men exceed.
- Line 259: the term 'situational' is not very common.
- Spell out scores when mentioned the first time.
- line 474: which kinds of VTE occurred?
- line 513: 'awarded' perhaps not suitable word.
- Line 578-587: relevant discussion on prediction models. Consider referring to the study: DOI: 10.1055/s-0038-1673330.
Author Response
Response Letter to Reviewers
Re: jcm-863082
We thank the reviewers for their thoughtful and detailed recommendations and believe their concerns can be satisfactorily addressed by the revised manuscript. In this letter, we describe changes made to address these concerns and provide explanatory text where required. Attached is a copy of the revised manuscript with revisions highlighted.
Regarding the points raised by reviewer 3:
The review "Prevention of Venous Thromboembolism in 2020 and Beyond" by Nicholson et al is a comprehensive work of VTE associated risk factors and prophylaxis.
The review is well-written and reads well. However, there are things to be discussed.
General considerations:
Given the fact that the manuscript is described as a review nothing is mentioned on how the literature was searched or evaluated. The review could definitely benefit from some more information on the methods used.
This work is not presented as a systematic review but rather as a narrative review, which are often published in other high impact factor journal without formal search strategies (NEJM, Blood etc). The breadth of this review and the experience of the authors has led to a combination of traditional and ad hoc literature review strategies being used at different parts of the review. We feel confident that between the collective experience of the authors and reviewers that we have minimized potential biases and omissions. Importantly, we support our statements with an exhaustive list of references.
As a reader the text could benefit from some reorganization of the headings. Before section 6, a heading stating "Risk factors for VTE" could be relevant. Also, I am confused about section 8. The text continues describing risk factors so why this middle section?
Unfortunately, the original intent of the section headings was lost, and “major” and “minor” section headings were merged before they were distributed to reviewers resulting in confusion.
Please let us know if the updated manuscript adds sufficient clarity to this issue. The intent is to separate intrinsic, patient-specific factors (which are traditionally not actionable/practical in isolation) and acquired, situational risk factors with their associated epidemiology, approach, and scoring systems.
The tables uses odds ratios. However, it is easier to interpret a HR or RR. Also, often the OR are not referred to in the text making the tables less useful. Not all risk factors are mentioned in the table, e.g. COVID19. The headings of the tables: could consider using the word 'association' instead of 'effect'.
The titles of both tables have been amended for precision of language (below)
We agree with the referee that measures of relative risk (such as RR or HR) are easier to interpret that odd ratios. However, the most common study designs (case control or cohort studies using logistic regression model) to examine the association between a risk factor and the occurrence of VTE, report the measure of association as an odd ratio (OR) rather than HR or RR. For this reason, this table reports OR. However, if the incidence of VTE < 10%, the OR would approximate RR or HR. So, in our case, reporting OR does not make a substantial difference in interpretation because in the context of primary prophylaxis, most populations have baseline risk < 10%.
We have reworded the title and substituted ‘effect’ with ‘association’.
In addition, we have provided new information for this review, which now discusses the association between immobility at work, travel, and testosterone supplementation which have not been studied as extensively (but anecdotally are the subject of significant interest to patients and practitioners).
Table 1. Association of Patient-Specific Risk Factors With Risk of First Episode of VTE [57–59].
|
INTRINSIC RISK FACTORS |
ODDS RATIO (OR) FOR VTE |
|
Elevated BMI >30 |
2.3 |
|
Heterozygous Prothrombin gene mutation |
2.8 |
|
Heterozygous Factor V Leiden gene mutation |
4.2 |
|
Homozygous Prothrombin Gene mutation |
6.7 |
|
Homozygous Factor V Leiden gene mutation |
11.5 |
|
Antithrombin Deficiency |
14.0 |
Table 2. Association of Acquired Risk Factors With The Risk of First Episode of VTE [73–78].
|
ACQUIRED RISK FACTORS |
ODDS RATIO (OR) FOR VTE |
|
Seated immobility at work* |
1.8 |
|
Long-Haul Travel† |
2.1 |
|
Testosterone supplementation |
2.3 |
|
Low risk COC (Levonogestrel) |
3.6 |
|
Pregnancy or Postpartum |
4.2 |
|
Trauma/Fracture |
4.6 |
|
Medical Hospitalization |
5.1 |
|
Neurologic Disease with Leg Paresis |
6.1 |
|
High risk COC (Desogestrel) |
6.8 |
|
Active Cancer |
14.6 |
|
Surgery |
21.7 |
Line 67-72: Could be relevant to discuss positive predictive values of diverse diagnostic tests and additionally cost/benefits of using the tests.
This point is well taken – We highlight the relevant issue with positive predictive values by mentioning that the incidence of VTE in those with suspected DVT or PE ranges from 5% to 20%. Consequently, when diagnostic tests are applied to a population with low to moderate VTE incidence, the positive predictive value is moderate (~60%) because false positives are common.
(Lines 68-76)
“Assessment of the impact of VTE on our health systems must also consider the burden of patients presenting to the emergency department with suspected VTE, as well as the burden of misdiagnosis. In the general population only one out of every five patients who are evaluated for VTE in the emergency department for VTE are diagnosed with VTE [24,25]. In pregnant patients only one in twenty-five patients evaluated for VTE are diagnosed with VTE [24,26]. Consequently, when diagnostic tests are applied to a population with low to moderate VTE incidence, the positive predictive value is moderate (~60%) and false positives are common. Because ruling out VTE often requires clinical evaluation, laboratory testing, and imaging studies the true healthcare burden of VTE must consider these costs as well as the cost of over and under diagnosis [27].”
These broad statements address the impact of underdiagnosis and overdiagnosis without creating too much overlap between this review and another in the same series. We prefer not to discuss individual diagnostic test performance and post-test probability as there is a separate article in this series that reviews testing and diagnosis of VTE in detail.
Line 95-98 and line 131: could discuss under-diagnosing of fatal PE. Historical autopsy studies all describe large proportion of un-recognised PE ante-mortem.
Historical studies detailing high rates of undiagnosed fatal PE were typically small series (Stein, Chest 1995), and occasionally thrombi that form postmortem have features that are difficult to distinguish from antemortem thrombi. A more recent series of >13,000 patients (https://www.ncbi.nlm.nih.gov/pmc/articles/PMC3654296/) reported a rate of 2.5% fatal PE. Underdiagnosis of PE remains a significant concern but may not be as consequential in the age of routine thromboprophylaxis as previously thought. Firm contemporary data in this area is lacking.
Given that uncertainty we examine what the role of discussing underdiagnosis of fatal PE may be in a manuscript on prevention of VTE and conclude that it would add weight to the importance of thromboprophylaxis. Our discussion of the benefits of thromboprophylaxis already strongly argues the merits of prevention over treatment (now lines 143-161). Thromboprophylaxis likely does decrease the rate of undiagnosed fatal PE, but this relationship is difficult to support with data, and we prefer to allow the existing arguments for prevention to stand as sufficient grounds for prevention without discussing undiagnosed PE.
Section 6: should be mentioned that the risk differentiates according to sex. At a younger age, women have the highest risk, later men exceed.
An important point. Added a paragraph in Section 6 (Now Section 5) to briefly describe the relationship between age and sex.
(Lines 223-227)
“Age exerts variable effects on the risk of VTE by sex. During childbearing years, the incidence of VTE increases in women and in the third decade of life the risk first VTE events in women outnumber those in men [16,43]. This effect is due to increased endogenous estrogen as well as the increased risk from introduction of exogenous hormonal therapies and pregnancy. Outside of childbearing years the incidence of VTE is greater in men [15,16]”
References
- Næss, I.A.; Christiansen, S.C.; Romundstad, P.; Cannegieter, S.C.; Rosendaal, F.R.; Hammerstrøm, J. Incidence and mortality of venous thrombosis: A population-based study. J. Thromb. Haemost. 2007, 5, 692–699, doi:10.1111/j.1538-7836.2007.02450.x.
- Silverstein, M.D.; Heit, J.A.; Mohr, D.N.; Petterson, T.M.; O’Fallon, W.M.; Melton, L.J. Trends in the incidence of deep vein thrombosis and pulmonary embolism: A 25-year population-based study. Arch. Intern. Med. 1998, 158, 585–593, doi:10.1001/archinte.158.6.585.
- Heit, J.A.; Spencer, F.A.; White, R.H. The epidemiology of venous thromboembolism. J. Thromb. Thrombolysis 2016, 41, 3–14, doi:10.1007/s11239-015-1311-6.
Section 7: nothing is mentioned on thrombophilia testing. Please deliberate this.
Have now included the term in the final paragraph of section 7 (now section 6) for clarity
Lines 268-281
“The ability to predict recurrent VTE is not substantially improved by thrombophilia testing, and VTE prophylaxis is not undertaken based on the detection of factor V Leiden, prothrombin gene mutation, or protein C or S deficiencies. The duration of anticoagulation therapy for patients with VTE should be determined on clinical grounds with attention paid to the circumstances and provoking factors of the inciting event rather than underlying genetic factors [71,72]. Small numbers of patients with homozygous factor V Leiden and compound heterozygous factor V Leiden and prothrombin gene mutation preclude definitive conclusions on the risk of recurrence. Conflicting reports exist in the literature as to whether these conditions lead to increased recurrence and are an indication for long-term anticoagulation [73–75]. Additionally, identification of thrombophilia does not appear to affect the efficacy of traditional anticoagulant agents, except for antithrombin deficiency which has the potential to interfere with the action of UFH or LMWH. Identifying asymptomatic genetic risk via thrombophilia testing only rarely affects clinical practice, and genetic studies are of clinical value primarily due to their strong interaction with certain acquired risk factors including estrogen therapy and pregnancy.”
Please be consistent on using either 'VTE' or 'venous thromboembolism'.
Changes to the section headings (below) should help to alleviate this problem somewhat. After defining the term VTE (Line 19) the use of VTE over Venous Thromboembolism should now be consistent with two exceptions:
- In headings (on guidance from https://blog.apastyle.org/apastyle/2015/10/an-abbreviations-faq.html)
- When the term is used to start a sentence it remains full. While the APA doesn’t specifically prohibit the use of acronyms to begin sentences, I find it somewhat abrasive and would prefer to leave the few examples of starting sentences with the full term as they are.
It is not consistent if the headings uses the term 'VTE' or if only the risk factor is mentioned (e.g. section 9 on hormonal therapy). I suggest not mentioning VTE in headings.
Removed the term VTE from nearly all section headings and used the term in full when it was required (i.e. Venous Thromboembolism used instead of VTE if it appears in a heading following guidance from the APA– this use is now limited to the opening sections). Headings that were changed:
- Approaches to Prevent Venous Thromboembolism
- Age is One of The Most Important Risk Factors
- Genetic Risk Factors
- Exogenous Hormonal Therapies
- Pregnancy
- Surgery
- Medical Hospitalization
- COVID-19 Infection
- A More Palatable Approach to Pharmacologic Prevention
Consider including a table on risk scores and associated purpose. Could also mention existing risk scores for VTE recurrence risk (DOI: 10.1055/s-0040-1708877)
Comparing systems for risk assessment is a challenging task due to the heterogeneous nature of their development, validation, and reporting. Figure 1 (below) has been developed to provide a high-level summary of scoring systems that are in use today, and appears after the first mention of a scoring system in the figure (Line 441)
Another article in this series looks at the risk of VTE recurrence in greater detail (https://doi.org/10.3390/jcm9051582), and apart from a brief discussion of recurrent VTE to assist with understanding the “hierarchy of clinical importance” we prefer not to discuss this important topic here.
When describing a new risk factor, please begin with describing the size of the problem. Section 13 line 464: when referring to a 'high rate' but nothing is described yet on these rates. Similar for other sections (e.g. section 14).
Regarding section 13 a significant amount of information was added about both hospitalized and ICU patients based on the input of another reviewer. This data requires contextualization due to risk of bias in these studies, and in this section specific data do not appear in the opening sentences for that reason. The revised section is as follows:
Lines (507-545)
“Data on VTE in asymptomatic, mild, and ambulatory COVID-19 patients are limited, and the risk of thrombosis in these settings is not well established. Early reports have shown the risk of VTE in hospitalized and severe COVID-19 infection is increased compared to patients with similar severity of illness who do not have COVID- 19. One series of COVID-19 patients in hospital but outside the ICU showed VTE incidence of 3.8%, an incidence of PE 2.5%, and a rate of positive imaging tests of 46% potentially indicating underdiagnosis in this population [148]. By comparison a study of hospitalized COVID-19 patients screened asymptomatic patients with D-dimer >1000ng/mL with whole leg compression ultrasound and found asymptomatic DVT in 14.7% of patients, but nearly all of these asymptomatic events were distal DVT [149].
Rates of VTE in COVID-19 patients requiring ICU care are even higher. An Early report from an ICU in Wuhan utilized screening ultrasound to show DVT occurred in 25% (20/81) of patients admitted with COVID-19 pneumonia [150]. Notably the baseline risk of VTE is lower in China, and thromboprophylaxis is not routinely used in these patients. Recent data reveals that increased rates of VTE are seen even with the use of prophylaxis. An ICU series in Italy reported 22% of patients developed VTE [151]. A publication from France compared the frequency of PE in 107 COVID-19 patients with a control period and with ICU admissions for influenza. In COVID-19 patients the frequency of pulmonary embolism was 20.6% compared to 6.1% in the control group and 7.5% in the group of patients with influenza [152]. In a study of two French intensive care units that utilized systematic screening with compression ultrasound 1-3 days after admission the proportion of patients with VTE was 69%, with 23% of patients diagnosed with pulmonary embolism despite prophylaxis, and in some cases despite therapeutic anticoagulation [153,154]. In a series of hospitalized patients in the Netherlands with only 37% of patients admitted to ICU the incidence of VTE was 11% (95% CI 5.8-17%) and 23% (95% CI 14-33%) at 7 days and 14 days, respectively [155]. With the addition of screening the overall VTE was 34% at 14 days despite weight-adjusted VTE prophylaxis with Nadroparin [155]. Follow-up data confirms a very high incidence of PE in this population – approximately 25% [146].
The development of VTE in COVID-19 patients appears to correlate with more severe disease course, higher D-dimer, higher CRP, and mortality [146,150,155]. High D-dimer >1.0 μg/mL and a Padua prediction score ≥4 may be useful markers for predicting DVT in hospitalized COVID-19 patients and require further study [156].
There are limitations and challenges with interpreting these early reports because of their mainly retrospective designs, short duration of follow-up, and consequent concerns about bias including publication and case ascertainment biases. Nonetheless, with a rate of VTE >20% in populations of severe COVID-19, many physicians are calling for increased doses of prophylactic anticoagulation or the empiric use of therapeutic dose anticoagulation [144,155,157,158]. However, intensifying anticoagulant therapy may increase the risk of bleeding, particularly in those who are critically ill. Ongoing randomized trials will determine whether these approaches will improve patient outcomes.”
Regarding section 14 additional text was added to include this information up front.
Lines 548-550
“In patients with cancer the risk of VTE is substantially increased due largely to tumor production of procoagulant and inflammatory substances which are released into the circulation. Rates of VTE in malignancy approximate 1% per year with significant variation based on tumor subtype from 0.5-20% per year [116].”
Reference Horsted 2012
Please be aware, that a proportion is not similar to a rate. A rate must always contain a time measure as opposed to a proportion (percentage) that is without a time scale. Example: line 480: proportions referred to as rates.
Some poor word choices in that section have now been edited. Revised text (Now line 523-529)
A publication from France compared the frequency of PE in 107 COVID-19 patients with a control period and with ICU admissions for influenza. In COVID-19 patients the frequency of pulmonary embolism was 20.6% compared to 6.1% in the control group and 7.5% in the group of patients with influenza [145]. In a study of two French intensive care units that utilized systematic screening with compression ultrasound 1-3 days after admission the proportion of patients with VTE was 69%, with 23% of patients diagnosed with pulmonary embolism despite prophylaxis, and in some cases despite therapeutic anticoagulation [146,147].
Prediction models: when describing prediction models and mentioning external validation please refer to performance measures of the models. Especially the calibration would be relevant in the VTE setting.
Figure 1 now provides an overview of whether internal and external validation have been successful in select prediction models. Measures of accuracy for these models are available through the original publications for the IMPROVE score and Khorana score and they have been included with the figure.
In addition, new text in the manuscript is now devoted discussing the validation of the CAPRINI and Khorana scores:
(Line 432-439)
“For individualized VTE prophylaxis after surgery the most widely used tool is the CAPRINI score. This scoring system has been validated across multiple surgical subtypes including critically ill surgical patients and is able to risk stratify patients for post-operative VTE [115–120]. Risk assessment models require patient-specific information to generate an estimate of that individual’s risk of VTE (see Figure 1). The CAPRINI score is best used with an online calculator and estimates an individual’s provides information about the post-operative risk of VTE over the first three months after surgery. This estimate can provide clinicians with guidance on which patients are at highest risk, and who may benefit from initial and extended pharmacologic prophylaxis after surgery.”
(Line 563-575)
“The Khorana score is the most widely known tool by which ambulatory cancer patients are risk stratified using clinical and laboratory criteria. Points are awarded assigned for a pre-chemotherapy platelet count ≥350, Hemoglobin level <10g/dL (or RBC growth factors), pre-chemotherapy leukocyte count >11x10⁹/L, and BMI ≥35kg/m². Pancreatic Cancer and stomach cancer each count for two points, and Lymphoma, Gynecologic, Bladder, and Testicular cancer count for one point. In the original retrospective and prospective validation cohorts, patients with a score ≥3 had a 7.1% and 6.7% risk of VTE at a median of 2.5 months, respectively [166]. Multiple prospective and series have confirmed the validity of the Khorana score and its use is endorsed by the most recent ASCO guidelines for VTE risk stratification in cancer [41,167,168]. The Khorana score is also valid for use in hospitalized cancer patients [169,170].Attempts to validate the Khorana score for specific subpopulations of cancer have been less successful, and in studies of lung cancer, hepatocellular carcinoma, acute myeloid leukemia, lymphoid malignancies the Khorana score did not adequately stratify or predict VTE events [171–174]”
Section 14: noting is mentioned on DOACs for cancer-associated VTE?
This revised version of the manuscript features expanded text on this highly relevant topic. (576-601)
“The recent CASSINI and AVERT studies used this risk stratification model to identify high risk patients who would benefit from DOAC thromboprophylaxis. These trials randomized patients with ambulatory cancers initiating chemotherapy with Khorana score ≥ 2 to placebo or prophylactic rivaroxaban or apixaban and respectively. The AVERT trial showed a significant 6% (4.2% vs 10.2%, HR 0.41, 95% CI 0.26-0.65, p < 0.001) absolute decrease in the rate of symptomatic and incidental VTE and an absolute increase in ISTH major bleeding of 1.7% (3.5% vs 1.8%, HR 2.00, 95% CI 1.01-3.95, p = 0.046) but no cases of fatal bleeding were observed [175]. Major bleeding events are more frequent in cancer patients, both due to the propensity of certain cancer subtypes to bleeding complications (including gastrointestinal and genitourinary cancers) and the tendency of cancer patients towards anemia and blood transfusion.
The CASSINI trial also enrolled patients with a Khorana score ≥ 2 but excluded those with primary or metastatic brain cancer. The composite primary end point of DVT, PE, and VTE-related death occurred in 5.95% of patients in the rivaroxaban group and 8.79% in the placebo group (HR 0.66, 95% CI 0.40-1.09, p = 0.10). The rates of major bleeding and fatal bleeding were also not statistically different between the study arms (HR 1.96, 95% CI 0.59-6.49, p = 0.265) with one fatal bleeding event observed in the rivaroxaban group [176]. Both CASSINI and AVERT were analyzed over a period of six months, but the mortality rates and drug discontinuation rates were much higher in CASSINI than AVERT, both of which may be attributable to more severe underlying disease in the CASSINI patients (Mortality approximately 20% CASSINI vs. 12% AVERT, drug discontinuation rate 47% CASSINI vs. 18% AVERT). Of the 62 patients who had a primary end-point event, 24 (39%) did so after stopping the study drug. A pre-specified intervention-period analysis in CASSINI, which considers only time on the drug, demonstrated a statistically significant reduction in the rate of VTE (2.6% vs. 6.4% HR 0.40 95% CI 0.20-0.80). Although the CASSINI trial did not meet its primary efficacy endpoint, the findings are consistent with those of AVERT. The high discontinuation rates of the DOACs over time in both trials and in treatment trials highlight the challenges with adherence to DOACs in the prevention and treatment of VTE in patients with cancer [177–179].”
Section 15 is too short. More could be said about future directions. It seems section 15 to 18 are not very well-balanced. More nuance is needed to strengthen the text in the end.
Unfortunately, the original intent of the section headings was lost, and “major” and “minor” section headings were merged before they were distributed to reviewers resulting in confusion. I’ve taken your suggestion to add a conclusion to summarize the findings of the review. Additionally, section 18 has been split to accommodate additional information on implementation challenges and machine learning separately to achieve greater nuance and balance when discussing these subjects.
Major heading III (formerly section 15) now includes additional information to make it clearer this paragraph introduces/signposts the sections that follow. (613-621)
“III. Future Directions for Prevention of Venous Thromboembolism
Advances in technology and pharmacology have already improved our ability to predict and prevent VTE. At least two major barriers to improving VTE prevention still exist. The first are the limitations on overall benefit to thromboprophylaxis inherent to all current anticoagulant medications. The second barrier is the imperfect science of individualizing VTE risk, and the inherent complexity of predicting multifactorial and competing phenomena. These final sections will discuss advances in pharmacology and technology including the importance of oral options for prevention, the potential of preventative options that do not cause bleeding, implementation of risk stratification strategies, and the potential of machine learning in future preventative efforts.”
Sections 15 and 16 are now split to discuss separate topics and allow more fulsome discussion of these topics (652-699)
“15. Implementing and Improving Scoring Systems
Efforts to predict and prevent venous thromboembolic disease are predicated on our ability to accurately identify patients at risk. The benefits of thromboprophylaxis must be weighed against the financial costs and potential for increased bleeding. Understanding which patients are at greatest risk can help medical practitioners and their patients to make intelligent decisions regarding the use of anticoagulants to prevent VTE. We must continue to collect and analyze large data sets on patient-specific and acquired risk factors, and how they interact to improve existing risk assessment models. Early successes with predicting VTE risk using biomarkers should prompt further research into the use of biomarkers in new situations. The discovery and evaluation of novel biomarkers to risk-stratify patients may be an area of significant interest going forward, and serum biomarkers such as Vascular Endothelial Growth Factor (VEGF), Interferon alpha (IFN- α), Interleukin-15 (IL-15), and Citrullinated histone H3 (H3cit) may see further investigation for this purpose [186–188].
Biomarkers are only one aspect of prediction. The development and validation of prediction models should seek to increase the accuracy of prediction without sacrificing usability [189,190]. As we seek to individualize preventative efforts, the primary obstacles that affect implementation of risk assessment models will be the complications and complexity of any proposed algorithm. Scoring systems and other risk assessment models can be implemented for use by health professionals if they are easily accessible and understood but developing sufficient predictive power for VTE often requires the combination of several variables. Scoring systems can quickly become time-consuming for use by physicians and hiring additional data entry personnel adds significant cost. Predictive algorithms that appropriately consider and calculate bleeding risks for individual patients to avoid harms from VTE prophylaxis increase complexity and add additional workload. Scoring systems developed in the current age must not lose sight of the practical challenges of implementation by treating clinicians.
- Better Prevention Through Technology
Electronic medical records (EMRs) have provided new opportunities for VTE prevention. Scoring systems and other decision support tools can be incorporated into EMRs for ease of access. Studies indicate computer alert interventions can increase adherence to appropriate VTE risk-stratification [165], reduce costs by avoiding unnecessary thromboprophylaxis in low-risk patients [166], and decrease preventable harm from VTE [126,127,166]. Implementation of such systems should consider whether the absolute benefit is worth the additional burden on healthcare providers as long as direct provider input is required for the system to function.
Smoother implementation of existing categorical scoring systems is laudable, but health technologies in the future will render such systems obsolete. By taking continuous variables such as age or weight and reducing them to binary or ternary variables, we necessarily sacrifice some predictive value. Predicting an individual’s risk of VTE is extremely challenging because no single variable is strongly predictive, and we are forced to rely on systems that incorporate multiple variables to produce meaningful predictive values for VTE. Machine learning has the capacity to overcome these challenges by recognizing patterns in complex sets of information that accurately predict the risk of VTE and bleeding [191]. When interpreted by such systems continuous variables does not need to be reduced to categorical variables for ease of use. Additionally, new clinical events can be fed into machine learning algorithms continuously, which in turn would allow such systems to adjust the weight of variables in risk calculations quickly and precisely.
Until such technology is available institutional protocols for VTE prophylaxis should be regularly reviewed to ensure they reflect recent studies detailing risks and benefits of this common medical intervention. Discussions at the institutional and individual level should seek to balance the complex relationship between the multifactorial risk factors for VTE and bleeding complications and strive for net clinical benefit in prescribing pharmacologic VTE prophylaxis.”
The text could benefit from a conclusion in the end to summarize the findings of the review.
This is an excellent suggestion. A conclusion has been added (701-714)
“IV. Conclusion
In the last half century, we have made tremendous progress in understanding the epidemiology and prevention of VTE. During that time we have moved from studies detailing the benefits of “mini-dose” unfractionated heparin given to post-operative patients to sophisticated algorithms that leverage patient-specific and acquired risk factors to determine which patients will derive the greatest benefit from VTE prophylaxis. Patients considering exogenous hormonal therapies or pregnancy can be accurately informed about their risk for VTE, and patients undergoing surgery, medical hospitalization, cancer treatments, and those with COVID-19 infection routinely receive preventative anticoagulant treatments when the benefits of such treatments are known to outweigh the risks. As scoring systems and other decision support tools increase in accuracy and complexity we risk overwhelming clinicians. In the future, novel biomarkers for predicting VTE, safer medications, and machine learning algorithms will revolutionize prediction and prevention of this common disorder. “
Minor comments:
please limit the use of abbreviations COC/OCP (not explained in table), HRT (line 292), ICU (line 462), ARDS (line 463).
Removed reference to OCP. Abbreviations of HRT and ARDS removed.
COC and ICU are defined and subsequently used frequently in their respective sections.
Line 37 and line 129: I agree that the worst consequence is fatal PE. However, it could be mentioned that preventing all types of VTE is important when using prophylaxis.
Agree. I’ve left the introductory paragraph as is and added to the paragraph at 129 to make this point clearer. (now lines 141-162)
“Preventing fatal PE is the primary goal of anticoagulant prophylaxis for VTE. The one-month case fatality rate for VTE ranges from 2.8-12% [15,21,42,43]. The case fatality rate from PE accounts for virtually all of the overall case fatality for VTE, and the initial presentation for 24% of PE patients is sudden death [44].
Prevention of VTE also avoids significant post-VTE morbidity. Three conditions that can develop despite appropriate treatment of VTE are post-thrombotic syndrome (PTS), chronic thromboembolic pulmonary hypertension (CTEPH), and post-PE syndrome. Post-thrombotic syndrome is a constellation of lower extremity symptoms that persist after treatment of DVT including pain, paresthesias, skin pigmentation, functional restriction, and rarely venous stasis ulcers. The incidence of PTS after VTE varies from ~20 to 60% (depending on the definition of PTS) at 1-2 years after VTE diagnosis with 4.0-5.6% of patients developing debilitating symptoms indicating severe PTS [45,46]. The prevalence of CTEPH after VTE varies significantly in the literature, and is estimated at 3-4% [47,48]. The development of CTEPH after PE is not affected by the intensity of anticoagulation after an index PE event, and likely represents a pathobiology distinct from acute PE [49]. Long-term impairments to gas exchange and right ventricular dysfunction are the subject of ongoing study under the label ‘Post-PE syndrome’. Post-PE syndrome encompasses several long-term functional deficits that can occur after acute PE and are associated with a reduced health-related quality of life [50,51]. The prevalence and potential severity of these conditions must be considered when determining the potential benefits of preventing VTE. Averting sudden death and reducing post-PE morbidity are not the only benefits of anticoagulant prophylaxis, and prevention of VTE is important to avoid patient discomfort, anticoagulant treatments and their associated risks, specialist visits, delays in procedures, and the potential for additional testing.”
Additional references are needed: line 94 (after 'mortality'), line 100 (after 'uncertain'), line 123 (the sentence starting with 'ongoing'), line 277 (after 'VTE'), 342 (after 'period'), line 375 (after 'surgery'), line 453 (after 'benefit').
Added references except line 375 – this first sentence of the paragraph was removed as it was too vague
Space is needed: line 39, line 540.
Fixed
Line 56: BMI should be abbreviated first time used
Awkward due to the use of brackets when BMI is first introduced– now spelled in entirety in line 34 to avoid confusion and abbreviation introduced in line 58
Line 76 "earlier presentation" is not similar to preventing VTE. Line 149: 'reduce the risk' not the same as 'prevent VTE'. Line 150: 'diagnosing early VTE' not the same as 'prevent VTE'.
76 and 150 I tend to agree with your statements but am unclear on what the suggestion is. Strategies which reduce the risk of death from PE are complementary to preventative efforts.
149 Whereas no available strategies are 100% effective at preventing VTE reducing the risk of VTE is a less aggressive way of stating the same goal.
Overall, I would argue efforts to reduce the mortality of PE, even if those efforts are inferior to true prevention are towards the same overarching goal, and not “out of bounds” for this review.
line 135: please clarify that PTS is a clinical diagnosis.
Revised text to include this point and better summarize PTS for readers:
Line 148-151
“Post-thrombotic syndrome is a clinical diagnosis and is composed of a variety of lower extremity symptoms that persist after treatment of DVT including pain, paresthesias, skin pigmentation, functional restriction, and rarely venous stasis ulcers”
Line 158: unsure what is meant by 'undertake active measures to prevent VTE'.'
Removed the word “active” to minimize confusion.
Line 174 (previously 158) now reads:
“The second approach is to undertake measures to prevent VTE.”
line 160: it would be relevant here to mention the uncertainties regarding duration of OAC and findings of extended treatment in medically ill, e.g. studies on Betrixaban.
Introducing these studies without a full, nuanced discussion runs the risk of confusing readers as extended prophylaxis is one of the more complex topics in VTE prevention. A fulsome understanding of risk stratification strategies, which have not been introduced prior to this point, should be a prerequisite for understanding the extended prophylaxis studies. If necessary, we are happy to insert more information about extended prophylaxis in later sections after risk stratification schemes are introduced.
Section 6: should be mentioned that the risk differentiates according to sex. At a younger age, women have the highest risk, later men exceed.
An important point now summarized in a new paragraph
Lines (224-229)
“Age exerts variable effects on the risk of VTE by sex. During childbearing years, the incidence of VTE increases in women and in the third decade of life the risk first VTE events in women outnumber those in men [16,43]. This effect is due to increased endogenous estrogen as well as the increased risk from introduction of exogenous hormonal therapies and pregnancy. Outside of childbearing years the incidence of VTE is greater in men [15,16].”
Line 259: the term 'situational' is not very common.
The term situational has been removed. The section heading now reads (Line 283):
“II. Prevention: Acquired Risk”
Spell out scores when mentioned the first time.
Added to CATS, IMPROVE (not applicable to CAPRINI, Khorana, Padua)
line 474: which kinds of VTE occurred?
Amended for clarity
“An Early report from an ICU in Wuhan utilized screening ultrasound to show DVT occurred in 25% (20/81) of patients admitted with COVID-19 pneumonia [126]”
line 513: 'awarded' perhaps not suitable word.
Agree. Revised paragraph as follows (564-569)
“Points are assigned for a pre-chemotherapy platelet count ≥350, Hemoglobin level <10g/dL (or RBC growth factors), pre-chemotherapy leukocyte count >11x10⁹/L, and BMI ≥35kg/m². Pancreatic Cancer and stomach cancer each count for two points, and Lymphoma, Gynecologic, Bladder, and Testicular cancer count for one point. In the original retrospective and prospective validation cohorts, patients with a score ≥3 had a 7.1% and 6.7% risk of VTE at a median of 2.5 months, respectively [139].”
Line 578-587: relevant discussion on prediction models. Consider referring to the study: DOI: 10.1055/s-0038-1673330.
This publication is valuable and highlights many of the same challenges we identified. Reference added at (667-668) alongside another reference (Cowley 2019)
“Biomarkers are only one aspect of prediction. The development and validation of prediction models should seek to increase the accuracy of prediction without sacrificing usability [189,190].”
We thank the reviewer for their comments.

Round 2
Reviewer 3 Report
Very detailed and relevant response-letter from the authors. Thank you for a well-written review.
Minor comment:
Title of figure 1 should be revised to 'Selected' instead of 'Select'.